# Hacking LM Arena via LLM Identification with Interpolated Preference Learning

## Abstract

Voting-based leaderboards, such as LM Arena, have become the predominant method for evaluating large language models (LLMs) on open-ended tasks, with their fairness fundamentally depending on the anonymity of model responses. While prior work has shown that simple statistical features can be used for LLM identification, these methods could be easily defended and lack the power to distinguish between stylistically similar models. To further investigate such a risk in more sophisticated ways, we introduce a model-driven LLM identification framework via learning from **I**nterpolated **pref**erence data (**I-PREF**). Our approach utilizes a triplet ranking loss to train the detector model, a process augmented with synthetic hard negatives generated via copy model fine-tuning and model interpolation. This strategy enables the detector to learn deep relational patterns beyond superficial statistics. We further enhance performance and stabilize training through adaptive and iterative curriculum learning. Experimental results show that I-PREF significantly outperforms the existing baselines, achieving improvements of about 30% in accuracy and 24% in AUROC.

## 1 Introduction

With the rapid emergence of Large Language Models (LLMs) (Touvron et al., 2023; Achiam et al., 2023; Team et al., 2023), a central challenge has arisen regarding how to evaluate their performance in a fair and reliable manner. Conventionally, the evaluation of LLMs has relied on fixed, answer-based benchmarks such as HumanEval (Chen et al., 2021) and MMLU (Hendrycks et al., 2020). However, this approach suffers from two major limitations: (1) it is difficult to adequately reflect the diversity of real-world usage scenarios (Liang et al., 2022), and (2) it struggles to fairly assess scenarios with multiple valid responses, as opposed to a single, definitive ground truth (Zheng et al., 2023). These limitations have led to a shift toward evaluation paradigms that incorporate user preferences (Stiennon et al., 2020; Zhou et al., 2023; Dubois et al., 2024), which have become the preeminent standard for assessing model quality in open-domain conversational contexts.

Within this context, voting-based leaderboards such as *LM Arena* (Zheng et al., 2023; Chiang et al., 2024) have emerged as a new axis of LLM evaluation. LM Arena adopts a pairwise comparison scheme in which users are presented with responses from two LLMs to the same prompt, shown anonymously, and their judgments are aggregated into leaderboard scores using Elo (Boubdir et al., 2024) or Bradley–Terry ranking (Bradley & Terry, 1952) algorithms. The core assumption of voting-based leaderboards is the *anonymity of model responses*, where individual judgments are presumed to depend solely on the content quality rather than the identity of the source model (Panickssery et al., 2024). If such anonymity assumption is violated through identification signals embedded in the responses, coordinated manipulation targeting specific LLM becomes feasible, thereby fundamentally undermining the credibility of the leaderboard (Singh et al., 2025).

Therefore, investigating the feasibility of LLM identification is critical for ensuring and improving the reliability of leaderboard-based evaluations, and this line of work has only begun to be explored (Min et al., 2025; Huang et al., 2025). For example, Huang et al. (2025) have shown that even simple statistical features, such as bag-of-words (BoW) or TF–IDF representations, can identify LLMs with high accuracy. However, manipulations based on detection with such simple statistical features can be easily mitigated, as defenders can down-weigh the influence of suspicious votes by exploiting the same statistical signals during leaderboard aggregation (LM Arena Team, 2024a;b).

Moreover, these methods often lack sufficient discriminative power, when two models exhibit only minimal differences at the surface level such as teacher and student models (Hinton et al., 2015) (see Table 2). While the limitations of simple statistical identification methods alleviate some concerns, they also raise a key research question: *whether more advanced identification techniques exist that could threaten the robustness and credibility of leaderboards?*

**Contribution.** In this paper, we propose a new model-driven LLM identification framework via learning from **I**nterpolated **pref**erence data (**I-PREF**). Our key idea is training a detector model specified for target LLM, which assigns higher score to target LLM's responses compared to other LLMs, using the collected data of queries and responses by multiple LLMs. As this approach depends on the scale of data that incurs the cost, we instead propose a novel approach to synthesize preference data via *model interpolation* (Wortsman et al., 2022; Ilharco et al., 2023); another model is trained to mimic the responses by target LLM, and then we construct the interpolated model using both original and fine-tuned models. Since this interpolated model inherently includes target LLM's knowledge, the responses from this model could serve as hard negative data for preference learning and also their difficulty could be explicitly controlled by adjusting the interpolation ratio. To better exploit this synthetic response, I-PREF trains the detector model using the inherent preference order within triplets of responses from <Target LLM (high), interpolated LLM (middle), and other LLMs responses (low)>. Specifically, we employ an adaptive curriculum learning that dynamically switches to doublet tasks when triplets fail, and progressively strengthen detection performance through iterative stage-wise training (see Figure 1 for the illustration).

To demonstrate the effectiveness of I-PREF, we conduct experiments across three state-of-the-art target LLMs (GPT-4o (OpenAI, 2024), Gemini-pro (Gemini Team et al., 2023), Claude4-sonnet (Anthropic, 2024b)) and two datasets (Alpaca (Taori et al., 2023) and Arena human preference (Tang et al., 2025)), comparing performance against four baseline methods. On average, I-PREF outperforms the existing LLM identification baselines by approximately 30% in Accuracy and 24% in AU-ROC across the three target LLMs and both datasets. Moreover, I-PREF also demonstrates strong identification performance when compared against family models (*e.g.*, Gemini-pro v.s. Gemini-flash), even in situations where the benchmark baselines fail to distinguish them. To better understand the underlying mechanism of I-PREF, we present a detailed analysis showing that it is robust and does not rely on any specific model backbone. In particular, we observe that during training the score distributions between target and non-target responses become increasingly separated, which highlights the growing discriminative capability of our approach.

## 2 RELATED WORKS

**Recent progress in LLM evaluation.** Early evaluations of LLMs primarily relied on static test sets and automated evaluation. Benchmarks such as GLUE (Wang et al., 2018), HumanEval (Chen et al., 2021), and MMLU (Hendrycks et al., 2020) used predefined questions to measure the LLMs' capabilities for general or specific tasks. Despite its simplicity and usefulness, these static benchmarks struggle to capture the open-ended generative abilities of modern LLMs and are vulnerable to issues such as test data contamination (Magar & Schwartz, 2022) and limited coverage of real-world user queries (Liang et al., 2022). To overcome these limitations, researchers have explored interactive evaluations that rely on preference judgments. One prominent approach is the LLM-as-a-judge paradigm, where a strong model is used to grade or rank the outputs of other models (Zheng et al., 2023). While this reduces reliance on human annotators, AI judges introduce their own biases—for example, favoring verbose outputs, preferring certain stylistic patterns, or being sensitive to prompt wording and positional effects (Zheng et al., 2023; Dubois et al., 2024). Consequently, directly integrating large-scale human preferences have emerged as a new alternative evaluation method. The most notable example is LM Arena (Chiang et al., 2024), where real users compare two anonymized model responses to the same prompt and vote for the preferred one. By aggregating millions of pairwise comparisons through Elo-style rating systems (Boubdir et al., 2024), LM Arena dynamically ranks models and provides evaluation signals that are more diverse and up-to-date than those of static benchmarks (Zheng et al., 2023).

**Vulnerabilities of crowdsourced, anonymous leaderboards.** Crowdsourced, anonymous platforms such as LM Arena were designed to promote fairness through human oversight, but their openness and anonymity also create new attack surfaces that adversaries can exploit (Chiang et al.,

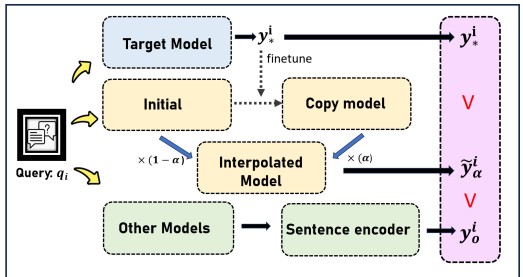 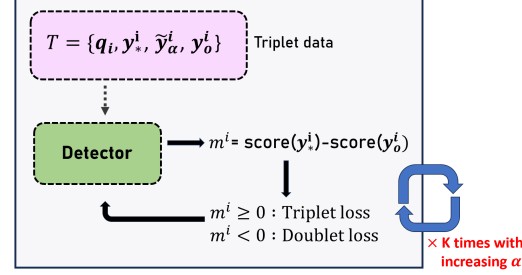

(a) Negative synthesis via model interpolation   (b) Adaptive and Iterative Curriculum learning

Figure 1: **Overview of I-PREF**. (a) *Hard negative synthesis via model interpolation*: we construct an interpolated model to generate synthetic hard negatives, which is constructed by combining the copy model trained to mimic the target LLM and its original backbone LLM under interpolation factor $\alpha$. Together with the target response and the most similar response selected from other models, these form triplet data to train detector model. (b) *Adaptive, iterative curriculum learning*: depending on margin between target and non-target responses and I-PREF reassigns samples to different tasks (doublet or triplet) accordingly. $\alpha$ is gradually increased, and this adaptive reassignment and training are iterated for $K$ stages, resulting in a progressively harder curriculum.

2024; Zheng et al., 2023; Zhao et al., 2024). Recent analyses warn that malicious voting can undermine leaderboard reliability even when human voters are involved (Zhao et al., 2024). The key factor behind this threat is the ability to de-anonymize model responses, *i.e.*, identify the source LLM by looking hidden feature in the response. Attackers can use a range of techniques, from simple frequency-based statistics such as BoW and TF–IDF to active watermarking and fingerprinting schemes, to identify the source model of a generated response with high accuracy (Kirchenbauer et al., 2023; Xu et al., 2024). Given such signals, adversaries can strategically cast a relatively small number of manipulated votes and thereby cause large shifts in ranking, as demonstrated in simulation studies (Min et al., 2025; Huang et al., 2025).

**Defender strategies and limitations.** As the threat of leaderboard manipulation emerges, platform operators have begun to deploy pragmatic defenses. For example, LM Arena employs techniques such as length/style normalization and sentiment control to diminish superficial cues of model identity, as well as bot-activity filters to defend against automated attacks (LM Arena Team, 2024b;a; Li et al., 2024). Although these defenses are effective in filtering out obvious statistical artifacts, they remain restricted to surface-level statistics. Our study shows that such safeguards are insufficient: even under these defenses, models can still be reliably identified even for closely related models (*e.g.*, GPT-4o vs. GPT-4o-mini). By explicitly uncovering these weaknesses, we demonstrate that current leaderboard mechanisms are far from complete and emphasize the need for more robust defense strategies to ensure their credibility as trustworthy benchmarks for LLM evaluation.

## 3 INTERPOLATED PREFERENCE LEARNING FOR LLM IDENTIFICATION

**Overview.** In this section, we present **I-PREF**, our model-driven approach for LLM identification, which learns to distinguish between the target LLM's responses and the responses generated by other LLMs through preference-based training. The key idea of I-PREF is training the detector model to yield higher reward for the target LLM's responses, by learning the preference over hard negative samples synthesized by interpolated model. We first establish the basic pair-wise preference learning approach using semantic similarity-based hard negative selection (Sec. 3.1). Next, we introduce our cost-efficient hard negative sample generation strategy through model interpolation (Sec 3.2). Lastly, we present an adaptive curriculum learning framework that dynamically adjusts between pair-wise and triplet training in Sec. 3.3. The overall illustration is presented in Figure 1.

**Problem setup.** Let us denote a set of accessible LLMs as $\mathcal{M} = \{M_1, M_2, \ldots, M_K\}$ where each LLM $M \in \mathcal{M}$ can generate response $y$ for the given input query $q$, *i.e.*, $y \sim M(q)$. Then, the goal of LLM identification is to find a detector model $f_\theta$ that assigns higher scores to response $y_*$ generated

by a target model $M_*$ compared to response $y_k$ from other models $M_k \in \mathcal{M} \setminus M_*$:

$$f_\theta(q, y_*) > f_\theta(q, y_k), \quad \forall M_k \neq M_* \tag{1}$$

To learn this detector model $f_\theta$, we assume that we have a set of queries $\mathcal{Q} = \{q^i\}_{i=1}^N$ and a set of responses $\mathcal{Y} = \{y_k^i | y_k^i \sim M_k(q^i), q^i \in Q\}$ generated by LLMs in $\mathcal{M}$. Then, one can learn the detector model $f_\theta$ using $\mathcal{Q}$ and $\mathcal{Y}$; for instance, simple regression model based on Bag-of-Words (Huang et al., 2025). However, these surface-level characteristics often fail to capture the subtle stylistic and reasoning patterns that distinguish modern LLMs. Therefore, I-PREF instead finetune pre-trained LLM to train detector model using $\mathcal{Q}$ and $\mathcal{Y}$ efficiently.

**Threat model.** We consider an attacker who is a model developer seeking to raise the ranking of their own model on a public, voting-based leaderboard such as LM Arena. The attacker participates as a normal user without access to system internals. As shown in (Zhao et al., 2024; Huang et al., 2025), strategic voting alone is sufficient to manipulate leaderboard outcomes. A crucial requirement for such manipulation is the ability to de-anonymize model responses —i.e., to infer which competing LLM produced each answer in a duel so that the attacker can cast selective votes that most benefit their target model. I-PREF directly enables this capability by accurately distinguishing multiple competing LLMs from their responses alone, without relying on metadata or privileged information.

### 3.1 LLM Identification via Pair-wise Preference Learning

To train the detection model for identifying target LLM, we first construct a dataset of preference pairs:

$$\mathcal{D} = \{(q^i, y_*^i, y_o^i)\}_{i=1}^N,$$

where for each query $q^i \in \mathcal{Q}$, $y_*^i$ represents the response from target model $M_*$, and $y_o^i$ represents the selected responses from other models $\mathcal{Y}^i = \{y_k^i | y_k^i \sim M_k(q^i), \forall M_k \neq M_*\}$. Specifically, to maximize discrimination difficulty during training, we select a hard negative $y_k^i$ as $y_o^i$ by computing semantic similarity between the target response and selecting the most similar one:

$$y_o^i = \arg\max_{y \in \mathcal{Y}^i} \; \mathrm{sim}\big(\mathbf{e}(y_*^i), \mathbf{e}(y)\big), \tag{2}$$

where $\mathbf{e}(\cdot)$ represents embeddings from the sentence encoder model,[1] and $\mathrm{sim}(\mathbf{u}, \mathbf{v})$ is a cosine similarity. Then, with $\mathcal{D}$, the detector model $f_\theta$ is trained to prefer target response $y_*^i$ over others' response $y_k^i$ by minimizing the following binary cross-entropy loss:

$$\mathcal{L}_{\mathtt{pair}} = -\frac{1}{|\mathcal{D}|} \sum_{(q^i, y_*^i, y_o^i) \in \mathcal{D}} \mathcal{L}_{\mathtt{pref}}(q^i, y_*^i, y_o^i), \quad \mathcal{L}_{\mathtt{pref}}(q, y_w, y_l) = \log \sigma\big(f_\theta(q, y_w) - f_\theta(q, y_l)\big) \tag{3}$$

This loss function enforces the preference modeling based on Bradley-Terry model (Bradley & Terry, 1952) by maximizing the probability that the target response receives a higher likelihood than the competing response.

For training, we use only the single hardest negative $y_o^i$. However, during validation and test, we expand the evaluation to include responses from all non-target LLMs. Namely, given $K$ LLMs, we evaluate against the remaining $K - 1$ LLMs (excluding the target) to comprehensively assess detector performance across the entire model space.

### 3.2 Cost-Efficient Hard Negative Generation via Model Interpolation

The promising way to improve the performance of detector $f_\theta$ is enlarging the size of $\mathcal{D}$ (*i.e.*, larger $N$) or expanding the diversity of $y_o^i$ by incorporating more LLMs (*i.e.*, larger $K$). However, this approach often incurs the additional costs for new data collection or API callings. This motivates our approach: *Can we generate synthetic hard negatives without incurring additional API costs while maintaining high discrimination difficulty?* To address this question, we propose a novel approach that synthesizes hard negatives motivated by model interpolation (Wortsman et al., 2022; Ilharco et al., 2023), rather than naively expanding the set of queries or the set of queried LLMs.

---

[1]We use `sentence-transformers/all-mpnet-base-v2`

Our approach begins by creating *copy model* $\hat{M}_*$ that is trained to mimic the response of target model $M_*$ through supervised fine-tuning:

$$\hat{M}_* = \arg\min_\phi \frac{1}{|\mathcal{D}|} \sum_{(q^i, y_*^i) \in \mathcal{D}} \mathcal{L}_{CE}(\hat{M}_\phi(q^i), y_*^i) \tag{4}$$

where $\mathcal{L}_{CE}$ denotes a standard language modeling cross-entropy loss. These copy model $\hat{M}_*$ serves as the foundation for generating synthetic responses by varying the levels of similarity compared to the target model $M_*$. Specifically, we synthesize responses of intermediate-difficulty from the model $\widetilde{M}_\alpha$ which is constructed with an interpolated model parameter $\widetilde{\phi}_\alpha$:

$$\widetilde{\phi}_\alpha = (1 - \alpha)\phi_{\texttt{init}} + \alpha\hat{\phi}_{\texttt{copy}}, \tag{5}$$

where $\alpha \in [0, 1]$ is hyperparameter to control difficulty, $\phi_{\texttt{init}}$ denotes the initial parameters of copy model, and $\hat{\phi}_{\texttt{copy}}$ denotes the parameters of $\hat{M}_*$, respectively. Using $\widetilde{M}_\alpha$, we construct the response-triplet dataset

$$\mathcal{T}_\alpha = \{(q^i, y_*^i, \widetilde{y}_\alpha^i, y_o^i)\}_{i=1}^N, \tag{6}$$

where $\widetilde{y}_\alpha^i = \widetilde{M}_\alpha(q^i)$ is the middle-quality response from interpolated model. Because weight-space interpolation induces smooth functional interpolation, the generations of $\widetilde{M}_\alpha$ vary between the initializer and the copy model: as $\alpha \to 0$ they resemble the initializer, as $\alpha \to 1$ they approach the target-like copy, and for $\alpha \in (0, 1)$ they yield **middle-difficulty negatives**, typically satisfying

$$0 < \Delta(q, \widetilde{y}_\alpha) < \Delta(q, y_o),$$

where $\Delta(q, y) = f_\theta(q, y_*) - f_\theta(q, y)$.

### 3.3 ADAPTIVE AND ITERATIVE CURRICULUM LEARNING

With the constructed response-triplet dataset $\mathcal{T}_\alpha$, we train our detector model $f_\theta$ by minimizing the following linear combination of binary cross-entropy losses:

$$\mathcal{L}_{\texttt{fin}} = \frac{1}{|\mathcal{T}_\alpha|} \sum_{i \in \mathcal{T}_\alpha} \mathcal{L}_{\texttt{trp}}^i, \ \mathcal{L}_{\texttt{trp}}^i = \lambda_1 \mathcal{L}_{\texttt{pref}}(q^i, y_*^i, \widetilde{y}_\alpha^i) + \lambda_2 \mathcal{L}_{\texttt{pref}}(q^i, \widetilde{y}_\alpha^i, y_o^i) + \lambda_3 \mathcal{L}_{\texttt{pref}}(q^i, y_*^i, y_o^i), \tag{7}$$

where $\lambda_1, \lambda_2, \lambda_3$ are hyperparameters that weigh the relative importance of each triplet component. While this triplet training is effective, some queries might suffer during training if they even fail to minimize a simple pair-wise loss (Eq. 3). To improve training by mitigating this issue, we consider adaptive curriculum learning. Specifically, for each query, we compute discrimination margin:

$$m^i = f_\theta(q^i, y_*^i) - f_\theta(q^i, y_o^i), \tag{8}$$

where $m^i < 0$ indicates that the detector $f_\theta$ currently fails to correctly discriminate target response $y_*^i$ and the hard negative response $y_o^i$. For those samples with $m^i < 0$, we apply $\mathcal{L}_{\texttt{pref}}$ instead of $\mathcal{L}_{\texttt{trp}}$; namely, focusing on solving easier problem.

Notable advantage of our framework is adaptability which allows to easily control the difficulty of training by varying the degree of interpolation $\alpha$. To further improve the performance of detection model $f_\theta$, we consider iterative curriculum learning that progressively increase discrimination difficulty. Specifically, we initially construct easier triplet data $\mathcal{T}_{\alpha_1}$ with small value of $\alpha_1$ and training the detection model following Eq. 7. After this stage, we construct a harder dataset $\mathcal{T}_{\alpha_2}$ with a larger value $\alpha_2 > \alpha_1$ and continue training. In this way, we perform iterative data construction and training across progressively harder stages, where we set $K = 2$ iterations in our experiments.

Overall, after the initial copy model training cost, our approach can generate unlimited hard negatives at various difficulty levels through parameter interpolation without additional API queries, achieving superior performance on challenging discrimination tasks.

## 4 EXPERIMENT

In this section, we conduct a comprehensive experiments to evaluate the proposed method, I-PREF, especially designed to address the following core research questions:

- **RQ1:** How effectively can I-PREF identify target LLMs compared to existing feature-based baselines? (Table 1, 2)

- **RQ2:** Do the components of I-PREF improve identification performance? (Table 3)

- **RQ3:** How data-efficient is I-PREF in achieving strong identification performance? (Figure 3)

- **RQ4:** Is I-PREF robust across different copy model and detector architectures? (Table 4)

## 4.1 SETUPS

**Datasets and metrics.** For the experiments, we consider two different sources for input queries: (1) *Alpaca dataset* (Taori et al., 2023), an instruction-following corpus of approximately 52k examples generated by LLM widely used for fine-tuning purposes and (2) *Arena human preference dataset* (Tang et al., 2025), a large-scale collection of over 140k pairwise human-labeled preferences obtained from real-world interactions in an LM arena setting. For the Alpaca dataset, we initiated a clustering process to read out 1,500 diverse samples and selected the final 1,400 for our query pool. For the Arena dataset, we filtered for single-turn, English-only conversations devoid of image inputs, randomly sampling 1,500 dialogues and retaining 1,400 after a cleaning process.

We then gather responses to these queries from a diverse set of 12 LLMs, which can be grouped into six families, each containing two variants: Llama-3.2-3B/Llama-3.1-8B (Grattafiori et al., 2024), Qwen2.5-3B/Qwen3-8B (Qwen Team, 2024b;a), Gemma-2-2B/Gemma-2-9B (Gemma Team et al., 2024), Claude-Sonnet-4/Claude3.5-Sonnet (Anthropic, 2024b;a), Gemini-Flash/Gemini-Pro (Gemini Team et al., 2023), and GPT-4o-mini/GPT-4o (OpenAI, 2024).

The resulting dataset, comprising query-response pairs, is pre-split into training (1,000), validation (200), and test (200) sets using a fixed random seed (seed=42) to ensure reproducibility and prevent data leakage. For training, we employ a sentence encoder to select the most semantically similar response from other models, which serves as the negative sample in triplet construction. Meanwhile, for validation and testing, we evaluate each query against all non-target models, resulting in 200 queries × 11 models = 2,200 scored pairs per split. Our main experimental setup focuses on three primary target LLMs: GPT-4o, Gemini-Pro, and Claude-4. For evaluation, we report both Accuracy and AUROC on the test sets by using validation sets for model selection. These metrics are computed and averaged across the 11 non-target LLMs to assess the generalization performance of our detector.

**Baselines.** We use feature-based detection methods trained with logistic regression as baselines (Huang et al., 2025). Specifically, we evaluate four types of statistical features proposed by Huang et al. (2025): TF-IDF (Term Frequency–Inverse Document Frequency), Bag-of-Words (BoW), and simple length-based features using word and character counts. For baselines, logistic regression models are trained on 1,000 training samples using five random seeds (100, 200, 300, 400, 500). The final baseline result is reported from the seed achieving the highest AUROC on the validation set, for the fairness in comparison with I-PREF that uses five epochs and validation processes.

**Implementation details.** We adopt GRM-Llama-3.2-3B[2], which is fine-tuned Llama-3.2-3B-Instruct for reward modeling as our primary detector backbone. Its training is performed with the AdamW optimizer ($\beta_1 = 0.9, \beta_2 = 0.999, \epsilon = 1 \times 10^{-8}$) and a weight decay of 0.01. We employ a cosine learning rate schedule with 100 warmup steps. The training process is divided into a two-stage iterative curriculum, with each stage running for 5 epochs. Stage 1 trains on easier triplets constructed with interpolated negatives at $\alpha_1 = 0.5$, using an initial learning rate of $1 \times 10^{-5}$. Stage 2 continues from the best Stage 1 checkpoint, training on harder triplets at $\alpha_2 = 0.75$ with a halved learning rate of $5 \times 10^{-6}$. We use a batch size of 2 with gradient accumulation of 4 (effective batch size 8) and FP16 mixed precision, leveraging DeepSpeed ZeRO-2 for memory efficiency. A weighted triplet loss is applied with coefficients $(\lambda_1, \lambda_2, \lambda_3) = (0.3, 0.3, 1.0)$. The final model checkpoint is selected based on the highest AUROC on the validation set.

For the copymodel, we fine-tune Phi-4-mini-instruct on the same pre-split datasets. The model is trained for 3 epochs with a batch size of 4 and a learning rate of $2 \times 10^{-5}$. The maximum sequence length is set to 512, and the best-performing checkpoint is selected based on the lowest validation loss. Further implementation details are provided in Appendix A.2.

---

[2]https://huggingface.co/Ray2333/GRM-Llama3.2-3B-rewardmodel-ft

Table 1: **Main results.** Target LLM identification performance (Average Accuracy and AUROC across 11 non-target LLMs) on two datasets: the Alpaca dataset and the Arena Human Preference dataset. I-PREF consistently outperforms feature-based baselines (TF–IDF, BoW, Length) across all three target LLMs (GPT-4o, Gemini-Pro, Claude-4).The best scores are highlited in **bold**.

| Target LLM | Method | Alpaca dataset | | Arena dataset | |
|---|---|---|---|---|---|
| | | Accuracy | AUROC | Accuracy | AUROC |
| GPT-4o | Length-Word | 0.493 | 0.551 | 0.611 | 0.545 |
| | Length-Char | 0.495 | 0.552 | 0.610 | 0.562 |
| | TF-IDF | 0.655 | 0.704 | 0.697 | 0.766 |
| | BoW | 0.631 | 0.692 | 0.709 | 0.772 |
| | I-PREF (Ours) | **0.887** | **0.875** | **0.908** | **0.920** |
| Gemini-Pro | Length-Word | 0.618 | 0.647 | 0.792 | 0.830 |
| | Length-Char | 0.618 | 0.639 | 0.768 | 0.815 |
| | TF-IDF | 0.780 | 0.851 | 0.844 | 0.926 |
| | BoW | 0.798 | 0.911 | 0.887 | 0.948 |
| | I-PREF (Ours) | **0.946** | **0.967** | **1.000** | **0.979** |
| Claude-4 | Length-Word | 0.525 | 0.529 | 0.631 | 0.621 |
| | Length-Char | 0.519 | 0.523 | 0.380 | 0.397 |
| | TF-IDF | 0.763 | 0.834 | 0.810 | 0.902 |
| | BoW | 0.767 | 0.861 | 0.851 | 0.929 |
| | I-PREF (Ours) | **0.982** | **0.954** | **0.946** | **0.976** |

Table 2: **Model-wise accuracy.** Cells highlighted in red indicate *family models* of the target high model (*e.g.*, GPT-4o vs GPT-4o-mini). The best scores are highlighted in **bold**. Abbreviations: C3.5 = Claude-3.5, C4 = Claude-4, G-F = Gemini-Flash, G-P = Gemini-Pro, G2-2B = Gemma-2-2B-it, G2-9B = Gemma-2-9B-it, GPT-M = GPT-4o-mini, GPT-4 = GPT-4o, L3.1-8B = Llama-3.1-8B-Instruct, L3.2-3B = Llama-3.2-3B-Instruct, Q2.5-3B = Qwen2.5-3B-Instruct, Q3-8B = Qwen3-8B.

| Target | Method | Compared LLMs | | | | | | | | | | | |
|---|---|---|---|---|---|---|---|---|---|---|---|---|---|
| | | C3.5 | C4 | G-F | G-P | G2-2B | G2-9B | GPT-M | GPT-4 | L3.1-8B | L3.2-3B | Q2.5-3B | Q3-8B |
| GPT-4o | Length-Word | 0.403 | 0.455 | 0.545 | 0.580 | 0.508 | 0.460 | 0.483 | – | 0.535 | 0.505 | 0.450 | 0.505 |
| | Length-Char | 0.405 | 0.468 | 0.543 | 0.570 | 0.510 | 0.470 | 0.483 | – | 0.530 | 0.508 | 0.445 | 0.513 |
| | TF-IDF | 0.718 | 0.723 | 0.683 | 0.720 | 0.678 | 0.678 | 0.568 | – | 0.618 | 0.605 | 0.563 | 0.658 |
| | BoW | 0.683 | 0.705 | 0.658 | 0.685 | 0.645 | 0.645 | 0.573 | – | 0.585 | 0.590 | 0.535 | 0.638 |
| | I-PREF (Ours) | **0.945** | **0.965** | **0.950** | **0.950** | **0.960** | **0.925** | **0.675** | – | **0.865** | **0.835** | **0.720** | **0.965** |
| Gemini-Pro | Length-Word | 0.690 | 0.695 | 0.538 | – | 0.595 | 0.658 | 0.613 | 0.625 | 0.573 | 0.593 | 0.645 | 0.578 |
| | Length-Char | 0.698 | 0.690 | 0.543 | – | 0.588 | 0.650 | 0.615 | 0.618 | 0.570 | 0.595 | 0.655 | 0.573 |
| | TF-IDF | 0.823 | 0.810 | 0.715 | – | 0.768 | 0.793 | 0.780 | 0.785 | 0.773 | 0.770 | 0.785 | 0.780 |
| | BoW | 0.818 | 0.813 | 0.763 | – | 0.823 | 0.820 | 0.778 | 0.780 | 0.795 | 0.805 | 0.798 | 0.790 |
| | I-PREF (Ours) | **0.965** | **0.955** | **0.940** | – | **0.950** | **0.955** | **0.925** | **0.915** | **0.945** | **0.955** | **0.940** | **0.965** |
| Claude-4 | Length-Word | 0.430 | – | 0.575 | 0.608 | 0.535 | 0.493 | 0.508 | 0.528 | 0.563 | 0.535 | 0.475 | 0.530 |
| | Length-Char | 0.435 | – | 0.570 | 0.600 | 0.540 | 0.495 | 0.513 | 0.525 | 0.563 | 0.535 | 0.470 | 0.538 |
| | TF-IDF | 0.575 | – | 0.768 | 0.798 | 0.793 | 0.815 | 0.760 | 0.768 | 0.810 | 0.793 | 0.758 | 0.753 |
| | BoW | 0.625 | – | 0.745 | 0.775 | 0.780 | 0.810 | 0.778 | 0.785 | 0.795 | 0.810 | 0.788 | 0.748 |
| | I-PREF (Ours) | **0.980** | – | **0.975** | **0.970** | **0.990** | **0.980** | **0.990** | **0.995** | **0.995** | **0.990** | **0.945** | **0.990** |

## 4.2 MAIN RESULTS

In Table 1, we present our main experimental results that report the LLM identification performance of I-PREF compared against feature-based baselines. Overall, I-PREF consistently surpasses all existing baselines in both AUROC and Accuracy, and this advantage holds across both the Alpaca and Arena Human Preference datasets. For example, I-PREF achieves an average Accuracy of 88.7% and AUROC of 0.875 to identify GPT-4o as the target model, clearly outperforming TF-IDF (65.5%) and BoW (63.1%). Such significant improvements are continuously observed for Gemini-Pro (Accuracy 94.6%, AUROC 0.967) and Claude-4 (Accuracy 98.2%, AUROC 0.954) as target LLMs, respectively. Notably, while I-PREF achieves strong gains on the Alpaca dataset, the improvements are even more pronounced on the Human dataset, which consists of real prompts collected from LM Arena. This suggests that I-PREF is not only effective on standardized benchmarks (Alpaca) but also exhibits higher discriminative power in realistic evaluation settings (Arena), indicating that detection accuracy may be even stronger in actual Arena deployments.

We further examine the robustness of I-PREF through model-wise breakdowns in Table 2. Remarkably, baselines with simple statistics such as TF-IDF and BoW struggle in within-family comparisons: for GPT-4o as the target, BoW achieves only 57.3% accuracy against GPT-4o-mini, while TF-

Table 3: **Ablation study.** Comparison of different configurations of I-PREF to train detector model. We vary the use of negative sampling strategy (Eq. 2), triplet loss (Eq. 7), adaptive curriculum (Eq. 8), and iterative training ($K =$1 or 2). We report average Accuracy and AUROC for three target LLMs (GPT-4o, Claude-4, Gemini-Pro), respectively. The best scores are highlighted in **bold**.

| Negative | Triplet | Iterative | Adaptive | GPT-4o | | Gemini-Pro | | Claude-4 | |
|---|---|---|---|---|---|---|---|---|---|
| | | | | Accuracy | AUROC | Accuracy | AUROC | Accuracy | AUROC |
| Hard | – | – | ✓ | 0.817 | 0.813 | 0.910 | 0.944 | 0.862 | 0.890 |
| Hard | ✓ | – | – | 0.849 | 0.858 | 0.951 | 0.961 | 0.952 | 0.933 |
| Hard | ✓ | – | ✓ | 0.845 | 0.868 | 0.936 | 0.966 | 0.973 | 0.945 |
| Easy | ✓ | – | ✓ | 0.821 | 0.852 | 0.916 | 0.948 | 0.897 | 0.887 |
| Random | ✓ | – | ✓ | 0.834 | 0.856 | **0.985** | 0.966 | 0.964 | 0.937 |
| Hard (Ours) | ✓ | ✓ | ✓ | **0.887** | **0.875** | 0.946 | **0.967** | **0.982** | **0.954** |

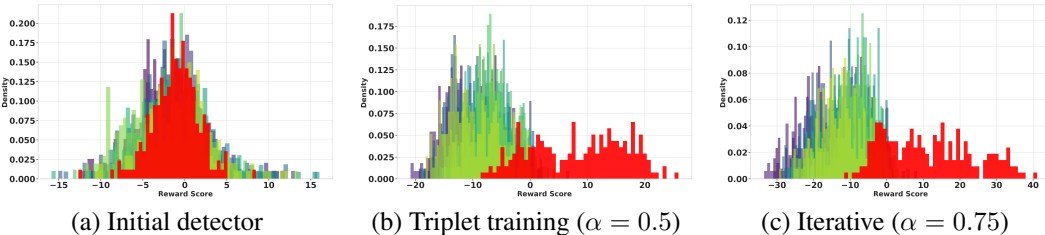

(a) Initial detector      (b) Triplet training ($\alpha = 0.5$)      (c) Iterative ($\alpha = 0.75$)

Figure 2: **Score distributions of the detector.** The detector is trained to identify Gemini-Pro. **Red** indicates the scores of the target model. (a) Initial detector before any fine-tuning. (b) After the first triplet training stage with interpolation factor $\alpha = 0.5$, where the detector begins to assign higher scores to the target model; the score range spans approximately from $-20$ to $20$. (c) After iterative curriculum training with progressively harder negatives ($\alpha = 0.75$), resulting in clearer separation between target and non-target models; the score range further expands from about $-30$ to $40$.

IDF records 56.8%, both far below our model-driven detector at 67.5%. A similar trend appears for Gemini-Pro, where BoW drops to 76.3% on Gemini-Flash compared to 94.0% with our method, and for Claude-4, where TF-IDF yields 57.5% on Claude-3.5 while our method reaches 98.0%. Specifically, length-based features (both words and characters) consistently record the lowest accuracy in nearly all comparisons, and in some cases even fall below 50.0%, indicating a complete failure of discrimination, as they perform worse than random guessing. These results show that baselines collapse when facing closely related family models, whereas I-PREF consistently maintains high accuracy, demonstrating robustness against confusion among distillation and family variants. We also provide more results of our model-wise breakdown studies in Appendix B.1.

### 4.3 MORE ANALYSES

**Ablation study.** Table 3 reports the effect of each component proposed in Sec. 3 for the identification performance of the trained detector model. First, comparing the pair-wise training (no synthesis by Sec. 3.2) with the triplet-based training shows that introducing the low response as an explicit negative improves AUROC by +0.055 on GPT-4o (0.813 → 0.868) and +0.055 on Claude-4 (0.890 → 0.945), confirming the benefit of triplet-based training. Second, adding the adaptive curriculum further improves performance: for GPT-4o, AUROC rises from 0.858 (no adaptive) to 0.868 (with adaptive), and for Claude-4 from 0.933 to 0.945. Third, incorporating iterative training yields the strongest overall results, pushing GPT-4o accuracy from 84.5% to 88.7% and Claude-4 from 97.3% to 98.2%. Finally, in terms of negative sampling, results show that hard negatives tend to provide the most robust improvements compared to easy or random choices. For example, on GPT-4o, Accuracy with hard negatives reaches 0.845, compared to 0.821 with easy and 0.834 with random, indicating that semantically challenging negatives are particularly beneficial for discrimination.

**Score distributions of trained detector.** To gain deeper insight into how such performance improvements occur, we visualize the evolution of score distributions assigned by the detector in Figure 2. At the initial stage (a) the detector does not exhibit any preference toward the target model, resulting in score distributions that resemble a nearly symmetric Gaussian-like shape with substantial overlap between the target and non-target models. In (b), after the first triplet training step with

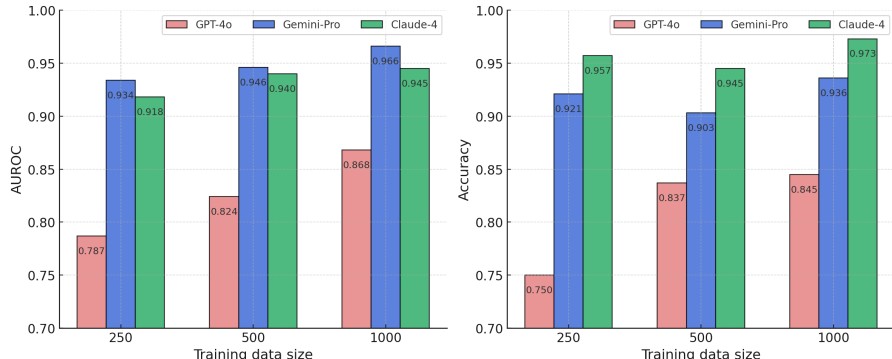

Figure 3: **Data scalability of I-PREF.** Performance trend with the proposed training framework as the number of training samples increases from 250 to 1000. Both Accuracy and AUROC show a generally linear improvement with larger training sets across all three target LLMs.

Table 4: **Robustness analysis.** Evaluation results with different copy model and detector backbones. We report average Accuracy and average AUROC when varying the underlying architectures.

| Component | Backbone | Target LLMs | Accuracy | AUROC |
|---|---|---|---|---|
| **Copy model** | Phi-4-mini | GPT-4o | 0.887 | 0.875 |
| | | Gemini-Pro | 0.946 | 0.967 |
| | | Claude-4 | 0.982 | 0.954 |
| | Qwen3-4B | GPT-4o | 0.839 | 0.845 |
| | | Gemini-Pro | 0.963 | 0.964 |
| | | Claude-4 | 0.991 | 0.953 |
| **Detector** | GRM-Llama-3.2-3B | GPT-4o | 0.887 | 0.875 |
| | | Gemini-Pro | 0.946 | 0.967 |
| | | Claude-4 | 0.982 | 0.954 |
| | Llama-3.2-3B | GPT-4o | 0.811 | 0.852 |
| | | Gemini-Pro | 0.921 | 0.964 |
| | | Claude-4 | 0.980 | 0.952 |

interpolation factor $\alpha = 0.5$, the distribution of the target model begins to separate from the others, indicating that the detector has started to assign systematically higher scores to the target responses. Finally, in (c), iterative curriculum training with increasingly harder negatives($\alpha = 0.75$) produces a clearer margin between the target and non-target distributions. This progression demonstrates that the I-PREF trained detector acquires stronger discriminative capability for identifying the target model, which directly explains the stronger performance on real-world data.

**Data scalability.** Figure 3 examines how detection performance scales with the amount of training data. It is observed that both AUROC and Accuracy show a generally increasing trend as the number of training samples grows from 250 to 1000, suggesting that additional data could further enhance performance. Nevertheless, with only 1000 instructions which corresponds to a modest API collection cost, detector using I-PREF already achieves strong results (average AUROC above 0.930 across three target LLMs). This demonstrates that I-PREF does not rely on prohibitively large datasets: even with a few hundred samples the detector remains competitive, and with 1000 samples it already provides robust accuracy in distinguishing state-of-the-art models.

**Robustness across models.** To address the question of whether I-PREF's performance depends on specific backbone choices, we evaluate robustness by varying both the copy model and detector architectures. When replacing the copy model backbone (from Phi-4-mini-instruct to Qwen3-4B ) or substituting the detector backbone (from GRM-Llama-3.2-3B to original Llama-3.2-3B), performance remains stable across all three target LLMs (Table 4). These findings suggest that I-PREF is not tied to a particular model architecture and can adapt robustly across different backbones.

## 5 CONCLUSION

In this paper, we presented I-PREF, a new model-driven approach to identify the source of LLM responses. I-PREF leverages triplet-based training with adaptive and iterative curriculum learning, supported by synthetic negatives generated via model interpolation. Extensive experimental results demonstrat that I-PREF consistently surpasses statistical baselines. Our findings highlight a critical vulnerability in current anonymous voting-based leaderboards, showing that despite pragmatic defenses, substantial loopholes remain exploitable. By explicitly uncovering these weaknesses, our work emphasizes the necessity of moving beyond statistical safeguards and provides a foundation for designing robust and trustworthy evaluation infrastructures for the next generation of LLMs.

## ETHICS STATEMENT

Our study explores the development of model-driven detectors designed to identify the source of responses from various LLMs. The primary motivation for this work is defensive: by improving the reliability of model identification, we can help secure preference-driven evaluation platforms like LM Arena and enhance the overall robustness of the LLM ecosystem (Bai et al., 2022; Team, 2024). This aligns with broader efforts in the AI safety community to design systems that better adhere to user intent and operational constraints, thereby fostering greater trust and predictability in AI systems (Askell et al., 2021; Ouyang et al., 2022). By highlighting the vulnerabilities of current evaluation systems to stylistic imitation, our work serves as a call to action for developing more resilient and trustworthy evaluation benchmarks.

Nevertheless, we recognize that technologies for model identification carry significant dual-use risks and ethical considerations. A primary concern is the potential for misuse in ways that undermine model privacy and intellectual property (Kirchenbauer et al., 2023; Xu et al., 2024; 2025). Powerful detectors could be used to de-anonymize models in competitive environments or enable unauthorized replication of proprietary model behaviors. Furthermore, there is a risk of an adversarial *cat-and-mouse* game, where malicious actors could leverage knowledge of these detection methods to refine their attacks, manipulating leaderboards or generating even more convincing disinformation that evades detection (Huang et al., 2025; Sadasivan et al., 2024). To mitigate these risks, we emphasize that our methods are intended for defensive research and should only be deployed within a comprehensive safety framework that includes robust content filtering, continuous monitoring, and transparent governance mechanisms (Le & et al., 2020; Inan et al., 2023).

## REPRODUCIBILITY STATEMENT

We provide implementation details (e.g., models, hyperparameters, and prompt design) and experiment setups (e.g., datasets and evaluation metrics) in Section 4.1 and Appendix A.2. The complete source code for our implementation and experiments will be made publicly available in a repository upon publication.

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

# A EXPERIMENTAL DETAILS

## A.1 MODEL SPECIFICATION

The models used in our dataset are as follows:

**Open-source models.**

- Meta / Llama-3.2-3B-Instruct[3]
- Meta / Llama-3.1-8B-Instruct[4]
- Alibaba / Qwen2.5-3B-Instruct[5]
- Alibaba / Qwen3-8B[6]
- Google / Gemma-2-2B-it[7]
- Google / Gemma-2-9B-it[8]
- Microsoft / Phi-4-mini-instruct

**API models.**

- Anthropic / Claude-4 Sonnet : `claude-sonnet-4-20250514`
- Anthropic / Claude-3.5 Sonnet : `claude-3-5-sonnet-20241022`
- Google / Gemini-Flash : `gemini-2.5-flash`
- Google / Gemini-Pro : `gemini-2.5-pro`
- OpenAI / GPT-4o : `gpt-4o`
- OpenAI / GPT-4o-mini : `gpt-4o-mini`

## A.2 TRAINING AND EVALUATION DETAILS

**Statistical baselines.** We evaluate four feature-based baselines trained with $\ell_2$-regularized logistic regression (SAGA solver, $C$=1.0), selecting the best seed by validation AUROC over five seeds (100, 200, 300, 400, 500).

- **Length–Word**: number of whitespace-separated tokens in a response; features are standardized.
- **Length–Char**: number of characters in a response; features are standardized.
- **BoW**: bag-of-words counts with n-grams up to 2 and `min_df`=2.
- **TF–IDF**: TF–IDF features with the same n-gram setting (1–2) and `min_df`=2.

For each baseline we train until convergence with a sufficiently large iteration cap (up to 2,000).

**Training details of I-PREF.** In our experiments based on a Qwen3-4B backbone, all parameter-interpolated variants tended to emit the special token "`<think>`" at the beginning of generations. To ensure comparability across models, we strip these scaffolding tokens and retain only the final answer content for both evaluation and training.

To construct the query–response corpus required for detector training, we generate responses for 1,400 queries from a diverse pool of 12 LLMs ($12 \times 1,400 = 16,800$ generations) using a unified decoding protocol. Local generations are produced with a high-throughput vLLM engine configured with sampling temperature 0.7, top-$p = 0.9$, a maximum of 4096 generated tokens, and stop sequences $\{$`<|eot_id|>`,`<|end_of_text|>`$\}$. For stability in cross-model comparisons, API-based generations are obtained with temperature fixed to 0 (greedy decoding).

---

[3] https://huggingface.co/meta-llama/Llama-3.2-3B-Instruct
[4] https://huggingface.co/meta-llama/Llama-3.1-8B-Instruct
[5] https://huggingface.co/Qwen/Qwen2.5-3B-Instruct
[6] https://huggingface.co/Qwen/Qwen3-8B
[7] https://huggingface.co/google/gemma-2-2b-it
[8] https://huggingface.co/google/gemma-2-9b-it

Table 5: **Model-wise AUROC results on Alpaca dataset.** Abbreviations: C3.5 = Claude-3.5, C4 = Claude-4, G-F = Gemini-Flash, G-P = Gemini-Pro, G2-2B = Gemma-2-2B-it, G2-9B = Gemma-2-9B-it, GPT-M = GPT-4o-mini, GPT-4 = GPT-4o, L3.1-8B = Llama-3.1-8B-Instruct, L3.2-3B = Llama-3.2-3B-Instruct, Q2.5-3B = Qwen2.5-3B-Instruct, Q3-8B = Qwen3-8B. Cells highlighted in red indicate *family models* of the target high model (e.g., GPT-4o vs GPT-4o-mini). The best scores are highlighted in **bold**.

| Target | Method | C3.5 | C4 | G-F | G-P | G2-2B | G2-9B | GPT-M | GPT-4 | L3.1-8B | L3.2-3B | Q2.5-3B | Q3-8B |
|---|---|---|---|---|---|---|---|---|---|---|---|---|---|
| GPT-4o | Length-Word | 0.472 | 0.509 | 0.620 | 0.675 | 0.570 | 0.507 | 0.514 | – | 0.584 | 0.551 | 0.480 | 0.584 |
| | Length-Char | 0.474 | 0.519 | 0.622 | 0.668 | 0.574 | 0.513 | 0.515 | – | 0.575 | 0.546 | 0.478 | 0.592 |
| | TF-IDF | 0.784 | 0.792 | 0.751 | 0.803 | 0.742 | 0.737 | 0.593 | – | 0.639 | 0.644 | 0.554 | 0.702 |
| | BoW | 0.763 | 0.790 | 0.741 | 0.792 | 0.713 | 0.710 | 0.613 | – | 0.627 | 0.642 | 0.533 | 0.685 |
| | I-PREF (Ours) | **0.918** | **0.940** | **0.940** | **0.942** | **0.941** | **0.942** | **0.921** | **0.693** | – | **0.841** | **0.844** | **0.705** | **0.903** |
| Gemini-Pro | Length-Word | 0.696 | 0.656 | 0.558 | – | 0.626 | 0.668 | 0.667 | 0.675 | 0.618 | 0.644 | 0.695 | 0.620 |
| | Length-Char | 0.688 | 0.643 | 0.547 | – | 0.615 | 0.659 | 0.659 | 0.668 | 0.616 | 0.640 | 0.691 | 0.603 |
| | TF-IDF | 0.892 | 0.876 | 0.766 | – | 0.846 | 0.873 | 0.864 | 0.863 | 0.837 | 0.843 | 0.850 | 0.855 |
| | BoW | 0.914 | 0.914 | 0.878 | – | 0.928 | 0.929 | 0.910 | 0.904 | 0.913 | 0.916 | 0.905 | 0.912 |
| | I-PREF (Ours) | **0.982** | **0.973** | **0.959** | – | **0.971** | **0.974** | **0.961** | **0.957** | **0.961** | **0.964** | **0.963** | **0.974** |
| Claude 4 | Length-Word | 0.423 | – | 0.591 | 0.656 | 0.567 | 0.492 | 0.493 | 0.491 | 0.578 | 0.531 | 0.436 | 0.564 |
| | Length-Char | 0.415 | – | 0.586 | 0.643 | 0.558 | 0.486 | 0.483 | 0.481 | 0.558 | 0.516 | 0.420 | 0.563 |
| | TF-IDF | 0.595 | – | 0.835 | 0.876 | 0.874 | 0.891 | 0.833 | 0.834 | 0.889 | 0.880 | 0.848 | 0.821 |
| | BoW | 0.701 | – | 0.822 | 0.863 | 0.897 | 0.922 | 0.875 | 0.887 | 0.888 | 0.899 | 0.887 | 0.832 |
| | I-PREF (Ours) | **0.934** | – | **0.953** | **0.980** | **0.968** | **0.946** | **0.954** | **0.967** | **0.963** | **0.957** | **0.914** | **0.954** |

Table 6: **Model-wise Accuracy on human preference dataset.** Abbreviations: C3.5 = Claude-3.5, C4 = Claude-4, G-F = Gemini-Flash, G-P = Gemini-Pro, G2-2B = Gemma-2-2B-it, G2-9B = Gemma-2-9B-it, GPT-M = GPT-4o-mini, GPT-4 = GPT-4o, L3.1-8B = Llama-3.1-8B-Instruct, L3.2-3B = Llama-3.2-3B-Instruct, Q2.5-3B = Qwen2.5-3B-Instruct, Q3-8B = Qwen3-8B. Cells highlighted in red indicate *family models* of the target high model (e.g., GPT-4o vs GPT-4o-mini). The best scores are highlighted in **bold**.

| Target | Method | C3.5 | C4 | G-F | G-P | G2-2B | G2-9B | GPT-M | GPT-4 | L3.1-8B | L3.2-3B | Q2.5-3B | Q3-8B |
|---|---|---|---|---|---|---|---|---|---|---|---|---|---|
| GPT-4o | Length-Word | 0.385 | 0.415 | 0.665 | 0.708 | 0.613 | 0.488 | 0.515 | – | 0.588 | 0.565 | 0.543 | 0.635 |
| | Length-Char | 0.395 | 0.428 | 0.675 | 0.713 | 0.620 | 0.508 | 0.513 | – | 0.588 | 0.575 | 0.555 | 0.633 |
| | TF-IDF | 0.788 | 0.778 | 0.700 | 0.725 | 0.690 | 0.695 | 0.645 | – | 0.665 | 0.663 | 0.585 | 0.733 |
| | BoW | 0.750 | 0.775 | 0.708 | 0.733 | 0.715 | 0.693 | 0.690 | – | 0.698 | 0.690 | 0.605 | 0.748 |
| | I-PREF (Ours) | **0.875** | **0.970** | **0.965** | **0.970** | **0.975** | **0.990** | **0.740** | – | **0.900** | **0.920** | **0.740** | **0.945** |
| Gemini-Pro | Length-Word | 0.835 | 0.793 | 0.585 | – | 0.830 | 0.858 | 0.848 | 0.813 | 0.785 | 0.805 | 0.825 | 0.733 |
| | Length-Char | 0.805 | 0.770 | 0.570 | – | 0.795 | 0.833 | 0.825 | 0.788 | 0.768 | 0.778 | 0.795 | 0.723 |
| | TF-IDF | 0.905 | 0.883 | 0.728 | – | 0.833 | 0.870 | 0.860 | 0.845 | 0.838 | 0.830 | 0.853 | 0.840 |
| | BoW | 0.890 | 0.885 | 0.818 | – | 0.903 | 0.913 | 0.895 | 0.898 | 0.863 | 0.900 | 0.905 | 0.885 |
| | I-PREF (Ours) | **1.000** | **1.000** | **1.000** | – | **1.000** | **1.000** | **1.000** | **1.000** | **1.000** | **1.000** | **1.000** | **1.000** |
| Claude 4 | Length-Word | 0.465 | – | 0.753 | 0.803 | 0.683 | 0.553 | 0.560 | 0.593 | 0.655 | 0.643 | 0.600 | 0.715 |
| | Length-Char | 0.538 | – | 0.255 | 0.205 | 0.350 | 0.468 | 0.475 | 0.428 | 0.370 | 0.378 | 0.408 | 0.305 |
| | TF-IDF | 0.583 | – | 0.818 | 0.845 | 0.865 | 0.840 | 0.850 | 0.825 | 0.820 | 0.830 | 0.823 | 0.818 |
| | BoW | 0.635 | – | 0.825 | 0.868 | 0.890 | 0.870 | 0.880 | 0.898 | 0.893 | 0.875 | 0.873 | 0.853 |
| | I-PREF (Ours) | **0.915** | – | **0.970** | **0.955** | **0.975** | **0.975** | **0.920** | **0.920** | **0.960** | **0.955** | **0.920** | **0.940** |

**Resource Details.** For the main development, we used a single NVIDIA RTX 6000 Ada 96GB GPU.

# B  DETAILED ANALYSES

## B.1  ADDITIONAL MODELWISE STUDIES

In Table 5, we report model-wise AUROC results on the Alpaca dataset. Beyond simple classification accuracy, AUROC provides a richer and threshold-independent view of separability, capturing how consistently the detector assigns higher scores to target responses compared to non-target ones. The consistently high AUROC values achieved by I-PREF across all non-target models confirm that our method offers not only strong accuracy but also robust and reliable discrimination, even in challenging settings such as family models.

In Table 6 and 7, we present results on the Human Preference dataset. Consistent with the Alpaca experiments, I-PREF achieves strong performance in both Accuracy and AUROC, showing that the detector remains effective across different sources of prompts. Notably, the improvements are particularly pronounced for family models (e.g., GPT-4o vs. GPT-4o-mini, Claude-4 vs. Claude-3.5), where traditional feature-based baselines tend to collapse. These findings confirm that across both

Table 7: **Model-wise AUROC on human preference dataset.** Abbreviations: C3.5 = Claude-3.5, C4 = Claude-4, G-F = Gemini-Flash, G-P = Gemini-Pro, G2-2B = Gemma-2-2B-it, G2-9B = Gemma-2-9B-it, GPT-M = GPT-4o-mini, GPT-4 = GPT-4o, L3.1-8B = Llama-3.1-8B-Instruct, L3.2-3B = Llama-3.2-3B-Instruct, Q2.5-3B = Qwen2.5-3B-Instruct, Q3-8B = Qwen3-8B. Cells highlighted in red indicate *family models* of the target high model (e.g., GPT-4o vs GPT-4o-mini). The best scores are highlighted in **bold**.

| Target | Method | C3.5 | C4 | G-F | G-P | G2-2B | G2-9B | GPT-M | GPT-4 | L3.1-8B | L3.2-3B | Q2.5-3B | Q3-8B |
|---|---|---|---|---|---|---|---|---|---|---|---|---|---|
| GPT-4o | Length-Word | 0.344 | 0.462 | 0.800 | 0.879 | 0.652 | 0.503 | 0.535 | – | 0.642 | 0.610 | 0.571 | 0.721 |
| | Length-Char | 0.351 | 0.479 | 0.795 | 0.867 | 0.650 | 0.513 | 0.534 | – | 0.632 | 0.602 | 0.576 | 0.706 |
| | TF-IDF | 0.866 | 0.883 | 0.780 | 0.808 | 0.749 | 0.755 | 0.713 | – | 0.716 | 0.719 | 0.616 | 0.817 |
| | BoW | 0.827 | 0.855 | 0.775 | 0.819 | 0.756 | 0.752 | **0.776** | – | 0.733 | 0.745 | 0.633 | 0.823 |
| | I-PREF (Ours) | **0.902** | **0.980** | **0.961** | **0.978** | **0.982** | **0.977** | 0.762 | – | **0.916** | **0.934** | **0.762** | **0.969** |
| Gemini-Pro | Length-Word | 0.910 | 0.845 | 0.565 | – | 0.844 | 0.885 | 0.879 | 0.860 | 0.834 | 0.844 | 0.858 | 0.806 |
| | Length-Char | 0.896 | 0.817 | 0.542 | – | 0.826 | 0.872 | 0.867 | 0.844 | 0.823 | 0.836 | 0.843 | 0.795 |
| | TF-IDF | 0.965 | 0.957 | 0.848 | – | 0.919 | 0.940 | 0.930 | 0.932 | 0.917 | 0.923 | 0.935 | 0.919 |
| | BoW | 0.952 | 0.937 | 0.905 | – | 0.955 | 0.959 | 0.957 | 0.958 | 0.944 | 0.955 | 0.957 | 0.948 |
| | I-PREF (Ours) | **0.983** | **0.987** | **0.972** | – | **0.984** | **0.985** | **0.975** | **0.975** | **0.976** | **0.977** | **0.973** | **0.984** |
| Claude 4 | Length-Word | 0.342 | – | 0.783 | 0.845 | 0.659 | 0.553 | 0.539 | 0.550 | 0.647 | 0.623 | 0.587 | 0.699 |
| | Length-Char | 0.662 | – | 0.235 | 0.183 | 0.357 | 0.463 | 0.479 | 0.467 | 0.374 | 0.402 | 0.426 | 0.324 |
| | TF-IDF | 0.660 | – | 0.908 | 0.947 | 0.950 | 0.931 | 0.937 | 0.934 | 0.913 | 0.910 | 0.925 | 0.906 |
| | BoW | 0.756 | – | 0.891 | 0.942 | 0.969 | 0.952 | 0.962 | 0.962 | 0.964 | 0.935 | 0.948 | 0.938 |
| | I-PREF (Ours) | **0.961** | – | **0.975** | **0.979** | **0.991** | **0.989** | **0.969** | **0.964** | **0.980** | **0.984** | **0.963** | **0.976** |

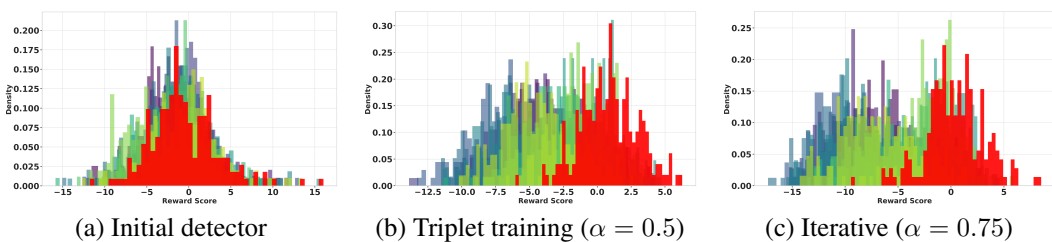

(a) Initial detector  (b) Triplet training ($\alpha = 0.5$)  (c) Iterative ($\alpha = 0.75$)

Figure 4: **Score distributions of the detector.** The detector is trained to identify GPT-4o as the target LLM. **Red** indicates the scores of the target model.

datasets and evaluation metrics, I-PREF provides a stable and reliable identification method that remains effective even under challenging conditions.

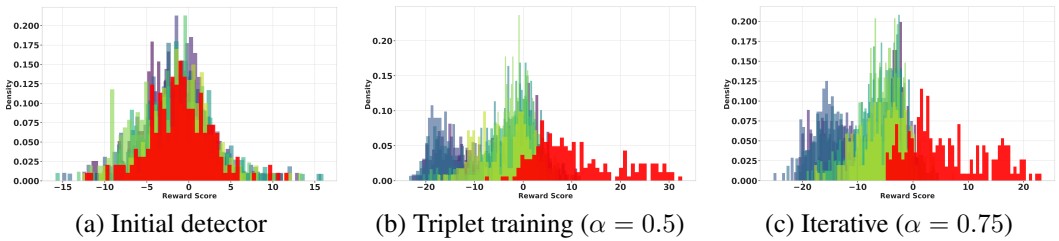

(a) Initial detector  (b) Triplet training ($\alpha = 0.5$)  (c) Iterative ($\alpha = 0.75$)

Figure 5: **Score distributions of the detector.** The detector is trained to identify Claude4 as the target LLM. **Red** indicates the scores of the target model.

## B.2 ADDITIONAL SCORE DISTRIBUTIONS

As shown in Figure 4, (a) the initial detector has no capability to distinguish the target model from the others, as their score distributions almost completely overlap. After the first triplet training stage with interpolation factor $\alpha = 0.5$ (b), the distribution of the target model (red) shifts toward the positive side, indicating that the detector begins to assign systematically higher scores to the target responses. Meanwhile, the non-target models' scores are pushed leftward, with their average moving toward approximately $-5.0$. Finally, under iterative curriculum training with $\alpha = 0.75$ (c), the separation becomes much clearer: the target distribution concentrates further on the positive axis, while the non-target models form a distinct peak around $-10$, reflecting the detector's strengthened discriminative capability.

Table 8: **Performance under a high-capacity reward model (Gemma-2-27B LoRA).** Using Phi-4-mini as the copy model, we evaluate Pairwise, Triplet, and Iterative training strategies across three target models. Increasing the reward model capacity preserves the relative performance ordering among strategies and yields consistent improvements.

| Target LLM | Accuracy | | | AUROC | | |
|---|---|---|---|---|---|---|
| | Pairsiwe | Triplet | Iterative | Pairwise | Triplet | Iterative |
| GPT-4o | 0.7197 | 0.7522 | 0.8055 | 0.7252 | 0.7256 | 0.7718 |
| Gemini-Pro | 0.8375 | 0.8618 | 0.9350 | 0.8241 | 0.8521 | 0.8947 |
| Claude-4 | 0.8545 | 0.8650 | 0.8868 | 0.8713 | 0.8722 | 0.8746 |

Moreover, as shown in Figure 5, (a) the initial detector again has no ability to distinguish the target model from the others, with the score distributions almost completely overlapping. After the first triplet training stage with interpolation factor $\alpha = 0.5$ (b), the detector starts to separate certain models such as the blue one, while the distinction between the target model (red) and another non-target (green) remains ambiguous. However, following iterative curriculum training with $\alpha = 0.75$ (c), the red, green, and blue distributions become much more clearly separated, demonstrating a substantially enhanced detection capability. Taken together, examining the score distributions across three different target models shows that I-PREF progressively stregthens the detector with stronger discriminative power, while emphasizing once more that such gains are achieved through nearly cost-free synthetic data generation.

## C USAGE OF AI ASSISTANTS

In preparing this work, we used AI-based writing assistants to improve sentence structure, correct grammatical errors, and enhance overall readability. These tools were employed soley for language refinement and did not contribute to the development of technical content, research methodology, or experimental analysis. All scientific ideas, results, and conclusions presented in paper were conceived and authored entirely by the researchers. The use of AI assistance was restricted to editorial purposes and did not affect the originality or intellectual contributions of the work.

## D ADDITIONAL EXPERIMENTS

### D.1 ALTERNATIVE REWARD MODEL ARCHITECTURES

In Table 8, we evaluate I-PREF using a high-capacity reward model (Gemma-2-27B[9] with LoRA fine-tuning) while fixing the copy model to Phi-4-mini. Across all three target models, Pairwise , Triplet, and Iterative training remain effective, and performance consistently improves with more sophisticated optimization schemes. These results indicate that increasing the reward-model scale does not break the method's comparative behavior and that I-PREF continues to benefit from iterative curriculum-based training even under higher-capacity discriminators.

### D.2 EFFECT OF COPY-MODEL CAPACITY ON IDENTIFICATION PERFORMANCE

In Table 9, we analyze the impact of copy-model capacity on identification performance by varying the size of the Qwen3 backbone from 0.6B to 8B parameters. We keep the evaluation setup identical to Section 4.1. Although larger copy models yield slightly higher scores, the performance gains diminish beyond a moderate scale. This trend is consistent with the objective of the copy model, which focuses on reproducing target-style characteristics rather than performing general-purpose language modeling; consequently, even lightweight models are sufficient to approximate the target style. These results suggest that increasing the capacity of the copy model offers limited utility for improving identification performance, while the detector remains the principal component responsible for discriminative modeling within I-PREF.

---

[9] https://huggingface.co/nicolinho/QRM-Gemma-2-27B

Table 9: **Effect of copy-model capacity on identification performance (Alpaca dataset).** We vary the capacity of Qwen3-based copy models from 0.6B to 8B parameters. While larger models yield slightly stronger performance, the gains diminish beyond a moderate scale, suggesting that copy-modeling, being a style-focused mimicry objective, does not require large capacity.

| Copy Model | GPT-4o | | Gemini-Pro | | Claude-4 | |
|---|---|---|---|---|---|---|
| | Accuracy | AUROC | Accuracy | AUROC | Accuracy | AUROC |
| Qwen3-0.6B | 0.826 | 0.819 | 0.941 | 0.947 | 0.938 | 0.942 |
| Qwen3-4B | 0.839 | 0.845 | 0.963 | 0.964 | 0.991 | 0.953 |
| Qwen3-8B | 0.878 | 0.824 | 0.949 | 0.963 | 0.954 | 0.955 |

Table 10: **Extended target model evaluation on open-source and closed-source LLMs (Alpaca dataset).** I-PREF maintains strong identification performance across both proprietary and fully open-source targets, demonstrating generalizability beyond closed-source models.

| Target LLM | Source Type | Accuracy | AUROC |
|---|---|---|---|
| GPT-4o | Closed | 0.887 | 0.875 |
| Gemini-Pro | Closed | 0.946 | 0.967 |
| Claude-4 | Closed | 0.982 | 0.954 |
| Llama-3.1-8B | Open | 0.901 | 0.897 |
| Qwen-3-8B | Open | 0.957 | 0.962 |
| Gemma-2-9B | Open | 0.905 | 0.901 |

## D.3 EVALUATION ON OPEN-SOURCE TARGET MODELS

In Table 10, we further extend our identification evaluation to open-source targets to verify whether I-PREF generalizes beyond proprietary frontier models. We consider three widely deployed open-source LLMs—Llama-3.1-8B, Qwen-3-8B, and Gemma-2-9B—and perform experiments under the same setup described in Section 4.1 using the Alpaca dataset. Across these models, I-PREF achieves an average Accuracy of 0.921 and an average AUROC of 0.920, demonstrating strong performance even when applied to models with different architectural origins and training pipelines. These results confirm that I-PREF is not limited to closed-source systems, but also generalizes to models commonly appearing in community-driven evaluation platforms, thereby enhancing its practicality in Arena-style environments.

## D.4 EFFECT OF INTERPOLATION STRENGTH ($\alpha$)

In Figure 6, we present a sensitivity analysis of the interpolation strength by varying the interpolation coefficient $\alpha$ across $\{0.1, 0.3, 0.5, 0.7, 0.9\}$ and evaluate the resulting detectors using both accuracy and AUROC of the resulting detectors. Across all models and all choices of $\alpha$, the performance curves remain largely stable. Although individual copy-models show slight variations in their preferred interpolation level, every interpolated configuration consistently outperforms the Pairwise baseline by a clear margin for both metrics.

Motivated by this robustness, we adopt simple and interpretable settings for our main experiments: $\alpha = 0.5$ for the single-step variant and $\alpha = 0.75$ for the iterative curriculum. While one could in principle tune $\alpha$ separately for each target model, our results suggest that such fine-grained optimization is unnecessary; the method delivers strong and reliable detection performance throughout the tested range. We therefore emphasize these values not as finely tuned hyperparameters, but as intuitive interpolation strengths that yield consistent gains over the non-interpolated Pairwise baseline.

## D.5 IDENTIFICATION ON MODELS WITH SIMILAR ELO RATINGS

To further examine whether I-PREF relies on substantial capability gaps between models, we additionally evaluate identification performance in a setting where all candidates exhibit highly similar

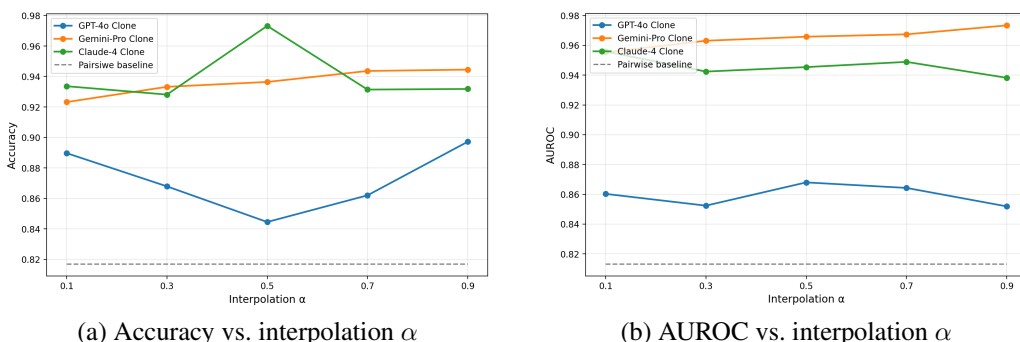

(a) Accuracy vs. interpolation $\alpha$          (b) AUROC vs. interpolation $\alpha$

Figure 6: **Effect of interpolation on detector performance.** Interpolating between the base and copy-model improves both (a) accuracy and (b) AUROC over the Pairwise baseline across all $\alpha$ values.

Table 11: **Model-wise accuracy among closely rated Arena models.** We evaluate I-PREF among models with highly similar Arena Elo scores, focusing on direct competitors rather than wide capability gaps.

| Target | GPT-4o | Gemini-Pro | Claude-4 | grok-4-fast | DeepSeek-v3.1 | glm-4.6 | kimi-k2 | qwen3-max |
|---|---|---|---|---|---|---|---|---|
| GPT-4o | – | 0.985 | 0.975 | 0.935 | 0.950 | 0.975 | 0.950 | 0.875 |
| Gemini-Pro | 0.885 | – | 0.890 | 0.935 | 0.675 | 0.565 | 0.835 | 0.910 |
| Claude-4 | 0.975 | 1.000 | – | 0.985 | 0.995 | 0.995 | 0.990 | 0.980 |

Arena Elo ratings.[10] Rather than focusing on wide disparities in model quality, this controlled setting reduces capability as a confounding factor and isolates stylistic and behavioral differences between models. We select a group of eight models clustered within a narrow Elo range on the LM Arena leaderboard: grok-4-fast, Gemini-Pro, DeepSeek-V3.1, glm-4.6, kimi-k2, GPT-4o, qwen3-max, and Claude-4. For each target model, we report identification performance against the remaining models in the group. As shown in Table 11, I-PREF continues to produce high and stable discrimination performance across all targets, even when capability differences are minimized. These results suggest that the effectiveness of I-PREF is not driven by trivial performance disparities between models; rather, the method captures model-specific stylistic and reasoning patterns that persist even among direct competitors with comparable leaderboard rankings.

### D.6 ADDITIONAL BASELINES FOR COMPARISON

To provide a comprehensive evaluation, we implemented two additional baseline methods that serve as strong comparison points for our proposed approach. Both baselines are evaluated on the Arena Human Preference dataset, with accuracy measured across all test samples.

**Neural-based multi-class classification baseline.** We implemented a neural network-based baseline in which a pre-trained transformer model is fine-tuned to classify the source of each generated response among all candidate LLMs following the source-attribution formulation of Tay et al. (2020). Given a pool of $N$ language models, the classifier is trained to predict which model produced a given answer. The training dataset comprises query-response pairs where each query is answered by all $N$ models, yielding $N$ labeled samples per query. We employ standard supervised learning with cross-entropy loss and select the best checkpoint based on validation accuracy.

For evaluation, we apply the model to pairs of candidate responses (e.g., a response from the target model and another from a competitor model) and compare the predicted logits. The response receiving a higher logit for the target class is considered to be from the target model. To reduce position bias, each test pair is evaluated in both forward and reversed order, and only consistent predictions are counted as correct.

---

[10] https://lmarena.ai/leaderboard/text

Table 12: **Accuracy of different identification baselines across three target models.**

| Method | GPT-4o | Gemini-Pro | Claude-4 | Average |
|---|---|---|---|---|
| LLM-judge | 0.679 | 0.620 | 0.631 | 0.643 |
| Neural-Based | 0.726 | 0.828 | 0.825 | 0.793 |
| **I-PREF (Ours)** | 0.908 | 1.000 | 0.946 | 0.951 |

**LLM-as-a-Judge baseline with few-Shot in-context learning.** As another strong baseline, we utilize an open-source LLM as an automated judge. This baseline leverages the in-context learning ability of large language models by providing several few-shot exemplars sampled from the training data. Each prompt consists of an instruction and two candidate responses (the target and a distractor), along with the ground-truth label indicating which response was produced by the target model.

For each test case, we query the judge model twice—once in each order (target vs. other and other vs. target)—while keeping the few-shot context fixed for all queries. The judge is instructed to identify the response most likely to have been generated by the target model. Only cases where the judge correctly identifies the target regardless of position are scored as correct; ambiguous or contradictory cases receive partial credit. To ensure computational efficiency, we utilize vLLM's prefix caching feature, as the lengthy few-shot prompt remains identical across all inference requests.

## E   SIMULATION FRAMEWORK FOR RANKING MANIPULATION IN LM ARENA

To quantify the practical impact of model-identification-based ranking manipulation, we perform extensive simulations using real-world chatbot arena data. Our goal is to estimate the computational cost—measured in interactions and votes—required to manipulate the rankings of top-tier models under realistic conditions.

**Simulation environment.** Our simulation setup follows the formulation by Huang et al. (2025), which models adversarial voting efficiency under Bradley–Terry and Elo ranking dynamics. We utilize the `lmarena-ai/arena-human-preference-140k` dataset released in July 2025,[11] which contains approximately 140,000 pairwise battles. Rankings are reconstructed with a Bradley–Terry (BT) model, and Elo scores are recalculated every 100 new votes. At each simulation step, the system draws two models uniformly at random from the candidate pool to form a synthetic battle. The attacker then receives two anonymized responses and chooses whether to cast a vote or abstain.

**Attacker objectives and metrics.** We define the attacker's objective as shifting a target model $M$ from its initial rank to a specified target rank (*e.g.*, Rank 1). We evaluate the feasibility of this attack using two key metrics:

- **Adversarial votes:** The number of votes actually cast by the attacker. This measures the direct influence exerted on the scoring system.
- **Total interactions:** The total number of queries submitted to the Arena. This metric represents the overall resource cost, as the attacker must spend time generating responses even for battles where they eventually abstain.

### E.1   SIMULATIONS WITH BASELINES

To evaluate the practical effectiveness of different model-identification strategies in a realistic Arena environment, we compare I-PREF against a broad set of baselines. Full descriptions of these baselines are provided in Sections A.2 and D.6. Under the simulation setup described above, our goal is to assess how many adversarial votes and total interactions are required to promote a target model upward in the leaderboard.

In this experiment, we consider `chatgpt-4o-latest-20250326`, which initially occupies Rank 5, and measure the effort needed to move it to Ranks 1–4. As shown in Table 13, all baselines

---

[11]https://news.lmarena.ai/opendata-july2025/

Table 13: **Required votes and interactions to promote `chatgpt-4o-latest-20250326`.** The target model starts from rank #5 and we report the adversarial votes and total interactions needed to move it to higher ranks under different detector accuracies.

| Target model = chatgpt-4o-latest-20250326 (current rank: #5) | Target rank: 1 (↑ 4) | Target rank: 2 (↑ 3) | Target rank: 3 (↑ 2) | Target rank: 4 (↑ 1) |
|---|---|---|---|---|
| LENGTH-CHAR | 1619 | 538 | 404 | 34 |
| LENGTH-WORD | 1616 | 560 | 394 | 36 |
| TF-IDF | 1630 | 563 | 394 | 36 |
| Neural-Based | 1604 | 526 | 448 | 36 |
| BoW | 1631 | 562 | 383 | 35 |
| LLM-judge | 1619 | 575 | 459 | 34 |
| I-PREF(Ours) | 1618 | 564 | 442 | 37 |

(a) # Votes

| Target model = chatgpt-4o-latest-20250326 (current rank: #5) | Target rank: 1 (↑ 4) | Target rank: 2 (↑ 3) | Target rank: 3 (↑ 2) | Target rank: 4 (↑ 1) |
|---|---|---|---|---|
| LENGTH-CHAR | 73600 | 24600 | 18200 | 1700 |
| LENGTH-WORD | 64500 | 21400 | 14600 | 1300 |
| TF-IDF | 55100 | 18500 | 12400 | 1100 |
| Neural-Based | 54500 | 18300 | 15900 | 1200 |
| BoW | 52300 | 17700 | 11600 | 1000 |
| LLM-judge | 63800 | 22200 | 18300 | 1500 |
| I-PREF(Ours) | 45800 | 16800 | 13000 | 1000 |

(b) # Interactions

require a similar number of direct adversarial votes to alter the model's ranking. However, the total number of interactions required to achieve the same rank shift varies substantially across detectors. In particular, methods with higher detection accuracy consistently reduce the number of required interactions, whereas weaker detectors incur significantly greater simulation cost.

This pattern highlights an important operational implication: the detector's accuracy directly controls how many Arena interactions an attacker must expend, and therefore determines the effective cost of executing a ranking-manipulation attack. Because I-PREF achieves the strongest detection performance among all evaluated baselines, it also achieves the lowest interaction cost across all target ranks, making ranking manipulation substantially more efficient relative to existing approaches.

### E.2 SIMULATIONS WITH I-PREF

Table 14: **Voting simulations on I-PREF.** The number of votes (a) and interactions (b) required to change the rankings of high-ranked models on the simulated leaderboard.Colors indicate direction of manipulation: upward promotion vs. downward suppression.

| Target model | Current rank | Target rank: 1 | Target rank: 2 | Target rank: 3 | Target rank: 4 | Target rank: 5 |
|---|---|---|---|---|---|---|
| gemini-2.5-pro | 1 | N/A | 11 | 409 | 507 | 856 |
| gemini-2.5-pro-preview-03-25 | 2 | 165 | N/A | 13 | 61 | 61 |
| grok-4-0709 | 3 | 218 | 25 | N/A | 52 | 56 |
| o3-2025-04-16 | 4 | 1791 | 639 | 466 | N/A | 5 |
| chatgpt-4o-latest-20250326 | 5 | 1615 | 489 | 439 | 37 | N/A |

(a) # Votes

| Target model | Current rank | Target rank: 1 | Target rank: 2 | Target rank: 3 | Target rank: 4 | Target rank: 5 |
|---|---|---|---|---|---|---|
| gemini-2.5-pro | 1 | N/A | 200 | 12100 | 14900 | 24700 |
| gemini-2.5-pro-preview-03-25 | 2 | 5000 | N/A | 500 | 2000 | 2000 |
| grok-4-0709 | 3 | 6400 | 600 | N/A | 1400 | 1500 |
| o3-2025-04-16 | 4 | 49300 | 18200 | 13400 | N/A | 100 |
| chatgpt-4o-latest-20250326 | 5 | 43800 | 13700 | 12400 | 1000 | N/A |

(b) # Interactions

We evaluate the effect of an adversary equipped with an I-PREF detector operating at its empirical accuracy on the Arena dataset (95.1%). When the target model appears in a battle, the detector attempts identification, and upon success, the attacker casts a strategic vote that promotes or suppresses the target depending on the objective.

The simulation operates under a passive sampling regime, as the target model appears in only a small fraction of battles due to uniform model sampling. Prior analyses report that the probability

of encountering a specific model in an Arena comparison is typically under 1% (Min et al., 2025; Huang et al., 2025). In our reconstructed leaderboard, the empirical participation rate is 3.3%, reflecting similar sparsity. Thus, the extent of manipulation is limited not by query count but by the stochastic availability of opportunities to act.

We quantify the effect of such adversarial interventions by measuring the number of adversarial votes required to shift model rankings. As shown in Table 14, demoting the top-ranked model from Rank 1 to Rank 2 requires only 11 downvotes, whereas promoting the second-ranked model to Rank 1 requires 165 positive votes. This asymmetry indicates that suppressing adjacent competitors is significantly more cost-efficient than exclusively boosting the target model's own score.

Overall, these findings demonstrate that a high-accuracy detector enables non-trivial leaderboard manipulation despite passive sampling constraints, and that adversarial influence is maximized when strategically directed toward down-ranking direct competitors rather than solely improving the target's own performance.

## E.3   SIMULATIONS WITH AGGRESSIVE I-PREF

Table 15: **Aggressive voting simulations on I-PREF.** The number of votes (a) and interactions (b) required to change the rankings of high-ranked models on the simulated leaderboard. Colors indicate direction of manipulation: upward promotion vs. downward suppression.

| Target model | Current rank | Target rank: 1 | Target rank: 2 | Target rank: 3 | Target rank: 4 | Target rank: 5 |
|---|---|---|---|---|---|---|
| gemini-2.5-pro | 1 | N/A | 66 | 628 | 1060 | 1371 |
| gemini-2.5-pro-preview-03-25 | 2 | 240 | N/A | 34 | 34 | 169 |
| grok-4-0709 | 3 | 315 | 34 | N/A | 56 | 168 |
| o3-2025-04-16 | 4 | 1188 | 250 | 225 | N/A | 78 |
| chatgpt-4o-latest-20250326 | 5 | 1169 | 244 | 196 | 64 | N/A |

(a) # Votes

| Target model | Current rank | # interactions | Target rank: 1 | Target rank: 2 | Target rank: 3 | Target rank: 4 |
|---|---|---|---|---|---|---|
| gemini-2.5-pro | 1 | N/A | 1900 | 15400 | 24000 | 26800 |
| gemini-2.5-pro-preview-03-25 | 2 | 3100 | N/A | 500 | 500 | 1900 |
| grok-4-0709 | 3 | 4100 | 300 | N/A | 500 | 1500 |
| o3-2025-04-16 | 4 | 14400 | 1900 | 1600 | N/A | 500 |
| chatgpt-4o-latest-20250326 | 5 | 14600 | 2000 | 1500 | 500 | N/A |

(b) # Interactions

In addition to the passive *do-nothing* strategy analyzed in Section E.2 , we consider a more powerful *aggressive* attack mode that exploits both the target model and its nearest competitors. Concretely, the simulator maintains an "enemy list" consisting of models whose current rank lies within a fixed window above the target (we use an aggressive range of 3 positions). Whenever the target appears in a battle and the detector successfully identifies it, the attacker behaves as before, casting a vote that promotes or suppresses the target depending on the objective. However, unlike the passive setting, the attacker can now also act in two additional cases: (i) if the target appears but detection fails, the adversary may still down-rank the opponent when it is on the enemy list; and (ii) even when the target does not participate in the battle, the attacker may cast a vote against a single enemy model if exactly one such competitor is present. The enemy list is updated periodically based on the current Bradley–Terry rankings, ensuring that the attack remains focused on the most relevant rivals as the leaderboard evolves.

Table 15 reports the number of adversarial votes and total interactions required to move each top-5 model between ranks under this aggressive policy. Compared to the passive case in Table 14, the required interaction budget decreases substantially across all scenarios. For example, promoting `o3-2025-04-16` from rank 4 to rank 1 previously required 49,300 interactions under the passive *do-nothing* attack, whereas the aggressive strategy achieves the same rank change with only 14,400 interactions. Similar reductions are observed for other models and objectives, indicating that leveraging both direct promotion of the target and suppression of nearby competitors yields a markedly more efficient manipulation strategy. These results suggest that, when combined with a high-accuracy detector such as I-PREF, vote-rigging policies that actively target the local neighborhood of the leaderboard can significantly amplify the practical impact of model identification on ranking outcomes.

