# OpenReview forum: "Hacking LM Arena via LLM Identification with Interpolated Preference Learning"
_ICLR.cc/2026/Conference — ICLR 2026 Conference Desk Rejected Submission_

### Official Review · Reviewer_xXHV · 2025-10-25

**Soundness:** 3
**Presentation:** 2
**Contribution:** 2
**Rating:** 4
**Confidence:** 5

**Summary:**

This paper introduces I-PREF (Interpolated Preference Learning), a novel framework for identifying the source of large language model (LLM) outputs. The work exposes a potential vulnerability in open evaluation platforms such as LM Arena, where anonymity is assumed in human preference voting.
I-PREF combines model interpolation (to synthesize hard negative samples) with triplet preference learning and adaptive curriculum training, enabling a detector to learn subtle stylistic cues of specific models. Experiments on datasets including the LM Arena human preference data show that I-PREF outperforms baseline methods (e.g., TF-IDF, BoW) by roughly +30% accuracy and +24% AUROC across multiple models (GPT-4o, Gemini-Pro, Claude-4). The findings highlight that even anonymized evaluation systems can be compromised through model-specific fingerprinting.

**Strengths:**

Innovative framework – Combines model interpolation and preference learning for LLM source detection.

Strong empirical performance – Significantly surpasses traditional feature-based baselines.

Data-efficient and low-cost – Requires only small datasets and no API access to the target model.

Robustness – Performs consistently across different architectures and settings.

**Weaknesses:**

The paper’s overall motivation is not entirely convincing. If the goal is to increase a model’s ranking or manipulate leaderboard outcomes, simpler techniques such as model watermarking could achieve this more directly without the need for a complex preference-learning framework. Moreover, the threat model remains vague throughout the paper. It is unclear whether the attacker is assumed to be a third-party observer attempting to de-anonymize model outputs, or a model developer seeking to improve their system’s ranking. This ambiguity weakens the justification and makes it difficult to assess the real-world relevance of the proposed attack scenario.


The interpolation parameter (α) is a key design choice in the proposed I-PREF framework, yet the paper does not include any ablation or sensitivity analysis on how varying α influences detection performance. While the authors describe a curriculum that gradually increases α, a quantitative study would be necessary to demonstrate robustness and justify the chosen configuration.

The authors argue that they train a copy model to avoid high API costs, implying access to GPU resources but limited API usage. In such a setup, it would be reasonable to leverage strong open-source models for local inference to obtain more diverse responses (but they did not do that). Although I understand that the target models (e.g., GPT-4o, Claude, Gemini) are closed-source, exploring additional open-source substitutes could have strengthened the experimental design and improved the generalizability of results.

**Questions:**

See weaknesses.

---

> ### Author Response · Authors · 2025-11-24
> **Response to Reviewer xXHV [1/4]**
>
> Dear Reviewer xXHV,
>
> We sincerely appreciate your thoughtful comments. We have carefully considered each of your questions and provide detailed responses below:
>
> ---
>
> **[W1] Clarification of motivation and why preference-learning framework rather than watermarking?**
>
> Thank you for raising this concern. While watermarking can in principle be used to verify a developer’s own model, it fundamentally enables only target-only manipulation. Because watermark verification requires possession of the secret key [1,3], an attacker can intervene solely in battles where their own model appears. However, prior analyses show that any given model participates in fewer than 1% of Arena comparisons [2,3], which forces watermark-based attacks into a passive regime where more than 99% of battles offer no opportunity to act. Under this limitation, even shifting a model upward by a single rank typically requires tens of thousands of positive votes, making watermark-only manipulation highly inefficient.
>
> In contrast, I-PREF provides a strictly more powerful capability: it recognizes not only the target model but also its nearby competitors. This enables strategic interventions based on downward suppression, where the attacker downvotes rival models even when the target is absent from the battle. To demonstrate this, we conducted new simulation experiments measuring the concrete impact an I-PREF–equipped adversary can exert on Elo-style rankings, following the setups of [2].
>
> As our simulation results below show, this capability dramatically increases the effectiveness of leaderboard manipulation. Under the passive setting (“do-nothing” when the target model is not sample), promoting the second-ranked model to Rank 1 requires 165 votes, whereas demoting the top-ranked competitor requires only 11 downvotes; it reveals a strong asymmetry to manipulate the ranking that watermarking cannot exploit.
>
>  (A) Votes
> | Target model                     | Current rank | Target rank: 1 | Target rank: 2 | Target rank: 3 | Target rank: 4 | Target rank: 5 |
> |----------------------------------|--------------|----------------|----------------|----------------|----------------|----------------|
> | gemini-2.5-pro                   | 1            | N/A            | ↓ 11          | ↓ 409          | ↓ 507          | ↓ 856          |
> | gemini-2.5-pro-preview-03-25     | 2            | ↑ 165          | N/A            | ↓ 13           | ↓ 61           | ↓ 61           |
> | grok-4-0709                      | 3            | ↑ 218          | ↑ 25           | N/A            | ↓ 52           | ↓ 56           |
> | o3-2025-04-16                    | 4            | ↑ 1791         | ↑ 639          | ↑ 466          | N/A            | ↓ 5            |
> | chatgpt-4o-latest-20250326       | 5            | ↑ 1615         | ↑ 489          | ↑ 439          | ↑ 37           | N/A            |
>
>  (B) Interactions
>
> | Target model                     | Current rank | Target rank: 1 | Target rank: 2 | Target rank: 3 | Target rank: 4 | Target rank: 5 |
> |----------------------------------|--------------|----------------|----------------|----------------|----------------|----------------|
> | gemini-2.5-pro                   | 1            | N/A            | ↓ 200         | ↓ 12100        | ↓ 14900        | ↓ 24700        |
> | gemini-2.5-pro-preview-03-25     | 2            | ↑ 5000         | N/A            | ↓ 500          | ↓ 2000         | ↓ 2000         |
> | grok-4-0709                      | 3            | ↑ 6400         | ↑ 600          | N/A            | ↓ 1400         | ↓ 1500         |
> | o3-2025-04-16                    | 4            | ↑ 49300        | ↑ 18200        | ↑ 13400        | N/A            | ↓ 100          |
> | chatgpt-4o-latest-20250326       | 5            | ↑ 43800        | ↑ 13700        | ↑ 12400        | ↑ 1000         | N/A            |
>
> Moreover, when the attacker employs an aggressive strategy that incorporates competitor suppression (i.e., downvoting high-ranking competitors), the number of interactions required to achieve the same rank changes drops substantially. For example, promoting o3-2025-04-16 from Rank 4 to Rank 1 requires 49,300 interactions under the passive target-only policy, but only 14,400 under the aggressive strategy.

---

> ### Author Response · Authors · 2025-11-24
> **Response to Reviewer xXHV [2/4]**
>
> (a) Votes
>
> | Target model                   | Current rank | Target rank: 1 | Target rank: 2 | Target rank: 3 | Target rank: 4 | Target rank: 5 |
> |--------------------------------|--------------|----------------|----------------|----------------|----------------|----------------|
> | gemini-2.5-pro                 | 1            | N/A            | ↓ 66           | ↓ 628          | ↓ 1060         | ↓ 1371         |
> | gemini-2.5-pro-preview-03-25   | 2            | ↑ 240          | N/A            | ↓ 34           | ↓ 34           | ↓ 169          |
> | grok-4-0709                    | 3            | ↑ 315          | ↑ 34           | N/A            | ↓ 56           | ↓ 168          |
> | o3-2025-04-16                  | 4            | ↑ 1188         | ↑ 250          | ↑ 225          | N/A            | ↓ 78           |
> | chatgpt-4o-latest-20250326     | 5            | ↑ 1169         | ↑ 244          | ↑ 196          | ↑ 64           | N/A            |
>
>
> (b) Interactions
>
> | Target model                   | Current rank | Target rank: 1 | Target rank: 2 | Target rank: 3 | Target rank: 4 | Target rank: 5 |
> |--------------------------------|--------------|----------------|----------------|----------------|----------------|----------------|
> | gemini-2.5-pro                 | 1            | N/A            | ↓ 1900         | ↓ 15400        | ↓ 24000        | ↓ 26800        |
> | gemini-2.5-pro-preview-03-25   | 2            | ↑ 3100         | N/A            | ↓ 500          | ↓ 500          | ↓ 1900         |
> | grok-4-0709                    | 3            | ↑ 4100         | ↑ 300          | N/A            | ↓ 500          | ↓ 1500         |
> | o3-2025-04-16                  | 4            | ↑ 14400        | ↑ 1900         | ↑ 1600         | N/A            | ↓ 500          |
> | chatgpt-4o-latest-20250326     | 5            | ↑ 14600        | ↑ 2000         | ↑ 1500         | ↑ 500          | N/A            |
>
> Overall, these results demonstrate that watermarking, which permits only target-only voting, represents the weakest and least effective manipulation strategy. I-PREF, by enabling recognition-driven suppression of competing models, unlocks a much larger space of adversarial actions and leads to significantly more efficient manipulation of Arena rankings. We have added these results and the corresponding discussion to Appendices E.2 and E.3 of the revised draft.
>
> ---
>
> **[W2]  The threat model remains vague throughout the paper**
>
> Thank you for pointing out this issue. In our work, the attacker is assumed to be a model developer attempting to improve the ranking of their own model on a public, voting-based leaderboard such as LMArena, rather than a passive third-party observer.
> This assumption is consistent with prior studies showing that leaderboard manipulation can be performed entirely through strategic voting without privileged access [1, 2]. The attacker participates in the platform as any normal user but with the intention to influence rankings.
>
> I-PREF directly supports this threat model by enabling reliable identification of multiple competing models (0.92 accuracy on Alpaca and 0.958 on Arena with only 1,000 samples). This allows the attacker to intervene strategically in a large fraction of battles and meaningfully affect leaderboard outcomes. We revise the draft to clarify this threat model and eliminate the ambiguity (lines 171-179)
>
> ---
>
> **[W3] Sensitivity analysis on how varying α influences detection performance**
>
> Thank you for raising this point. To address your concern, we conducted an ablation study in which we varied the interpolation coefficient α across five settings (0.1, 0.3, 0.5, 0.7, 0.9) and evaluated I-PREF using both AUROC and accuracy. Across all values of α, the interpolated detectors consistently outperform the Pairwise-setting by a clear margin, demonstrating the benefit of incorporating interpolation into the representation space.
>
>
> (a) Accuracy vs. interpolation α
>
> | α    | GPT-4o | Gemini  | Claude-4 | Pairwise |
> |------|--------|----------|-----------|----------|
> | 0.1  | 0.8897 | 0.9232   | 0.9336    | 0.8168   |
> | 0.3  | 0.8679 | 0.9332   | 0.9281    | 0.8168   |
> | 0.5  | 0.8445 | 0.9364   | 0.9732    | 0.8168   |
> | 0.7  | 0.8620 | 0.9436   | 0.9314    | 0.8168   |
> | 0.9  | 0.8972 | 0.9445   | 0.9318    | 0.8168   |
>
> (b) AUROC vs. interpolation α
>
> | α    | GPT-4o | Gemini  | Claude-4 | Pairwise |
> |------|--------|----------|-----------|----------|
> | 0.1  | 0.8603 | 0.9553   | 0.9559    | 0.8131   |
> | 0.3  | 0.8524 | 0.9631   | 0.9424    | 0.8131   |
> | 0.5  | 0.8680 | 0.9658   | 0.9454    | 0.8131   |
> | 0.7  | 0.8643 | 0.9674   | 0.9489    | 0.8131   |
> | 0.9  | 0.8519 | 0.9734   | 0.9382    | 0.8131   |

---

> ### Author Response · Authors · 2025-11-24
> **Response to Reviewer xXHV [3/4]**
>
> While individual target models show small performance differences depending on the choice of α, the overall trends indicate that I-PREF is not highly sensitive to the exact interpolation strength. This reliable performance throughout the tested range further demonstrates our choice not to introduce an additional hyperparameter search for α. Instead, we selected α=0.5 for the single-step variant and α=0.75 for the iterative curriculum simply as these values are intuitive and reflect straightforward interpolation settings, rather than being heavily optimized for any specific model.
> We have added these results and the corresponding discussion to Appendix D.4 of the revised draft.
>
> ---
>
> **[W4] Strong open-source models for local inference to obtain more diverse responses**
>
>
> Thank you for the thoughtful question. To address your comment, we conducted additional experiments using stronger open-source models. Our findings indicate that the copy model in I-PREF does not need to be large, as its role is specifically to approximate the target model’s stylistic tendencies rather than to perform general-purpose language modeling. As a result, scaling up the copy model produces only marginal improvements, and strong open-source generators are not necessary for our framework.
>
> In detail, we evaluated copy models ranging from 0.6B to 8B parameters while keeping the rest of the identification pipeline fixed. As shown in the table below, performance increases only marginally as the copy model grows, with diminishing returns beyond a moderate scale. Even the smallest model already captures the target model’s stylistic cues sufficiently well for the detector to operate effectively. This aligns with the intended function of the copy model, which is to approximate the target’s response style, not to expand the diversity or reasoning depth of responses.
>
>
> | Copy Model   | GPT-4o Accuracy | GPT-4o AUROC | Gemini-Pro Accuracy | Gemini-Pro AUROC | Claude-4 Accuracy | Claude-4 AUROC |
> |--------------|------------------|--------------|----------------------|-------------------|--------------------|-----------------|
> | Qwen3-0.6B   | 0.826            | 0.819        | 0.941                | 0.947             | 0.938              | 0.942           |
> | Qwen3-4B     | 0.839            | 0.845        | 0.963                | 0.964             | 0.991              | 0.953           |
> | Qwen3-8B     | 0.878            | 0.824        | 0.949                | 0.963             | 0.954              | 0.955           |
>
> Furthermore, our experiments confirm that the detector is the primary discriminative component in I-PREF. Identification accuracy is largely driven by the detector’s ability to exploit interpolated hard negatives, whereas increasing the generative capacity of the copy model provides limited additional benefit. In other words, computational investment in the detector, not in a larger copy generator, is what meaningfully improves identification performance. We have added these results and the related discussion to Appendix D.2 of the revised draft.
>
> ---
>
> **[W5] Exploring additional open-source substitutes could have strengthened the experimental design and improved the generalizability of results**
>
>
> Thank you for the constructive suggestion. We agree that evaluating open-source target models is important for demonstrating generalizability. To address this, we additionally evaluated I-PREF on the three open-source models already included in our experimental setup (Llama-3.1-8B, Qwen-3-8B, and Gemma-2-9B), using the same identification protocol as in the main experiments. The results are presented below
>
>
> ### **Extended target model evaluation on open-source and closed-source LLMs (Alpaca dataset)**
>
> | Target LLM     | Source Type | Accuracy | AUROC |
> |----------------|-------------|----------|-------|
> | GPT-4o         | Closed      | 0.887    | 0.875 |
> | Gemini-Pro     | Closed      | 0.946    | 0.967 |
> | Claude-4       | Closed      | 0.982    | 0.954 |
> | Llama-3.1-8B   | Open        | 0.901    | 0.897 |
> | Qwen-3-8B      | Open        | 0.957    | 0.962 |
> | Gemma-2-9B     | Open        | 0.905    | 0.901 |
>
> Here, I-PREF maintains strong performance even when the target model is fully open-source, achieving consistently high Accuracy and AUROC across Llama-3.1-8B, Qwen-3-8B, and Gemma-2-9B. Importantly, I-PREF performs well despite these models having different architectures, training pipelines, and stylistic characteristics compared to closed-source frontier models. This confirms that the effectiveness of our method does not rely on proprietary behaviors of models, but instead stems from the robustness of our preference-based detection framework.

---

> ### Author Response · Authors · 2025-11-24
> **Response to Reviewer xXHV [4/4]**
>
> Overall, these evaluations on open-source models show that I-PREF generalizes beyond closed-source models and remains effective in community-driven, Arena-style environments where open-source models frequently appear as direct competitors. We have added these results and the corresponding discussion to Appendix D.3 of the revised draft.
>
> ---
>
>
> [1] Kirchenbauer et al., A Watermark for Large Language Models, 2023.
> [2] Huang et al., Exploring and Mitigating Adversarial Manipulation of Voting-Based Leaderboards, 2025.
> [3] Min et al., Improving Your Model Ranking on Chatbot Arena by Vote Rigging, ICML 2025.
>
>
> ---
> If you have any further questions/concerns, please do not hesitate to let us know.
>
> Thank you very much,
> Authors

---

### Official Review · Reviewer_QwZn · 2025-10-29

**Soundness:** 3
**Presentation:** 3
**Contribution:** 3
**Rating:** 4
**Confidence:** 4

**Summary:**

This paper studies the risk that voting-based, anonymity-preserving leaderboards (e.g., LM Arena) can be de-anonymized — i.e., that a voter (or attacker) can identify which model produced a response and then use that ability to manipulate leaderboard outcomes. The authors propose I-PREF, a model-driven identification pipeline that trains a detector to prefer responses from a specific target LLM. Key ideas are: (1) constructing hard negatives by fine-tuning a copy model to mimic the target and then interpolating model weights to produce intermediate (“middle-difficulty”) responses; (2) triplet preference learning (target > interpolant > other) with a weighted triplet loss; and (3) adaptive, iterative curriculum learning that switches between triplet and pairwise losses and progressively increases interpolation difficulty. Experiments on two datasets (Alpaca and an Arena human-preference sample) and 12 LLMs (open and API) show I-PREF substantially outperforms simple statistical baselines (BoW, TF-IDF, length) on Accuracy and AUROC, including within-family distinctions (e.g., GPT-4o vs GPT-4o-mini). Ablations confirm benefits of triplet loss, adaptive curriculum, and iterative training. The paper discusses dual-use risks and frames the work as defensive research intended to inform more robust leaderboard defenses.

**Strengths:**

* Clear, practical method. The interpolated negative synthesis is an economical way to produce graded hard negatives without additional API calls; combined with triplet learning and adaptive curriculum it is conceptually simple and effective.

* Strong empirical gains over surface baselines. Across multiple target LLMs and two datasets, I-PREF shows large improvements in Accuracy/AUROC (e.g., average Accuracy up to ≈0.887–0.982 for different targets) and is robust in within-family comparisons where TF-IDF / BoW fail. Results and ablations (Tables 1–4, Table 3 ablation) support the claims.

**Weaknesses:**

* Baselines are too narrow. Comparisons are limited to simple feature-based detectors (TF-IDF, BoW, length). Stronger baselines (fine-tuned detectors trained on more sophisticated features, instruction-fingerprinting methods, watermark/fingerprint detectors, or recent instruction-fingerprinting papers) are missing; this makes it hard to position I-PREF’s advantage relative to state-of-the-art identification defenses/attacks.

* Limited evaluation on defender countermeasures. The paper notes LM Arena style/length normalization defenses but does not evaluate whether those or more aggressive countermeasures (e.g., paraphrasing, stochastic decoding, automated post-processing) degrade I-PREF effectiveness. Empirical tests of plausible defenses would strengthen the defensive framing.

**Questions:**

N/A

---

> ### Author Response · Authors · 2025-11-24
> **Response to Reviewer QwZn**
>
> Dear Reviewer QwZn,
>
> We sincerely appreciate your thoughtful comments. We have carefully considered each of your questions and provide detailed responses below:
>
> ---
>
> **[W1] Stronger baselines from the literature**
>
> Thank you for pointing out the missing baselines. To address this concern, we have incorporated substantially stronger baselines for model source identification beyond simple statistical methods. Specifically, we implemented (1) a neural multi-class transformer classifier trained to attribute each response to one of the candidate models [1], and (2) a strong LLM-as-a-judge baseline equipped with few-shot examples of identification cases. As watermark/fingerprinting methods necessitate the access to models which is not possible with our closed-source target models, we omit those in the consideration.  Detailed implementation descriptions are provided in Appendix D.6, and the corresponding results are summarized below:
>
>
> | Method        | GPT-4o | Gemini-Pro | Claude-4 | Average |
> |---------------|--------|------------|----------|---------|
> | LLM-judge     | 0.679  | 0.620      | 0.631    | 0.643   |
> | Neural-Based  | 0.726  | 0.828      | 0.825    | 0.793   |
> | I-PREF (Ours) | 0.908 | 1.000 | 0.946 | 0.951 |
>
>
> As shown in the table, both the neural classifier and the LLM-as-a-judge baseline struggle to reliably discriminate among high-performing models such as GPT-4o, Gemini-Pro, and Claude-4, achieving accuracies in the 0.62–0.83 range. This reflects the inherent difficulty of the identification task due to the high stylistic and structural similarity across modern frontier LLMs.
> In contrast, I-PREF achieves significantly higher accuracy across all target models (0.908–1.000), with an average accuracy of 0.951, outperforming both neural and LLM-based attribution methods by a large margin. These results demonstrate that even when compared against significantly stronger and more sophisticated baselines, I-PREF provides the most reliable and consistent source-identification performance. We will incorporate these new baselines into the main tables of the final draft.
>
> ---
>
> **[W2] Limited evaluation on defender countermeasures. Empirical tests of plausible defenses would strengthen the defensive framing.**
>
>
> What we aim to clarify in this work is that Style Control does not meaningfully weaken I-PREF, because it only normalizes surface-level stylistic features while our method relies on deeper behavioral signals that remain intact
> Style Control in LM Arena adjusts leaderboard scores after voting by removing the influence of simple stylistic attributes such as length, formatting, and markdown usage [2,3]. This makes it effective against manipulation strategies that depend on shallow, easily detectable cues. However, I-PREF does not rely on these surface-level patterns. Instead, it captures model-driven generative characteristics that originate from the model’s internal generation process. These signals are not removed by Style Control, which operates only on predefined, human-interpretable stylistic norms.
>
> Among the more aggressive countermeasures that have been suggested (e.g., paraphrasing, stochastic decoding, or automated post-processing), we note that these transformations fundamentally alter the content or semantics of the model’s output, making them unsuitable as practical countermeasures in an Arena setting where the platform is designed to evaluate the original behavior of each model. Applying such transformations would distort the model’s answers and make it unclear whether changes in identification accuracy arise from genuine robustness or simply from degrading the model’s response quality.
>
> For these reasons, our analysis focuses on realistic defensive mechanisms that preserve the original generation behavior of the model. Under such conditions, the behavioral signals exploited by I-PREF remain stable and continue to support reliable identification, even when stylistic normalization is applied. Moreover, statistical baselines that depend on length, formatting, or other shallow distributional cues are indeed neutralized by Style Control, whereas the deeper behavioral patterns captured by I-PREF are unaffected by such normalization allowing our method to maintain consistently superior performance even under these defenses.
>
> ---
>
> [1] Yi Tay et al., Reverse Engineering Configurations of Neural Text Generation Models, ACL 2020
> [2] https://lmsys.org/blog/2024-08-28-style-control/
> [3] https://colab.research.google.com/drive/19VPOril2FjCX34lJoo7qn4r6adgKLioY?ref=news.lmarena.ai#scrollTo=C4xnVybEy0OO
>
> ---
>
> We hope our responses resolve the points you raised. If you have any further questions, please don't hesitate and let us know.
>
> Thank you very much,
> Authors

---

> > ### Comment · Reviewer_QwZn · 2025-11-24
> >
> > Thanks for the author's response. The added experiment and explanation addressed my concerns and I have updated my score accordingly.

---

> ### Author Response · Authors · 2025-11-25
> **Response to Reviewer QwZn**
>
> Dear Reviewer QwZn,
>
>
> We are happy to hear that our rebuttal successfully address your concern and appreciate for raising the score! Please don’t hesitate if you have any further question.
>
> Sincerely,
> Authors

---

### Official Review · Reviewer_PG4o · 2025-10-31

**Soundness:** 3
**Presentation:** 3
**Contribution:** 2
**Rating:** 4
**Confidence:** 2

**Summary:**

This paper introduces I-PREF, a training framework for LLM source attribution that targets the identification of a specific model from its responses. The method builds a detector by first training a copy model to imitate the target, then generating intermediate hard negatives via weight interpolation between the copy and its initialization. The detector is optimized with triplet preference learning that contrasts the target response, the interpolated negative, and the most similar non-target response selected by embeddings. An adaptive curriculum switches between pairwise and triplet objectives based on the margin, and an iterative schedule increases difficulty by raising the interpolation coefficient. The aim is to enable practical de-anonymization of LLM outputs and to assess potential risks to rating systems such as LM Arena.

Experimentally, the paper evaluates across 12 models from multiple families and reports consistent gains over feature-based baselines such as length, BoW, and TF IDF with logistic regression. The study includes per-target accuracy and AUROC, ablations on interpolation and curriculum, backbone swaps that show transfer, and analyses of family-level cases where closely related variants are harder to separate. The results suggest that identity cues persist beyond shallow features, and the authors discuss implications for evaluation security and mitigation.

**Strengths:**

1. **Significance of the Problem.** The paper addresses a timely research problem. Anonymous, voting-based leaderboards like LM Arena are a primary standard for evaluating SOTA models, and this paper investigates a critical security vulnerability in their core assumption of anonymity.

2. **A New Detector Framework.** The authors propose I-PREF, a well-reasoned framework for training an identification detector. The strategy of synthesizing hard negatives via model interpolation is a cost-effective solution. When combined with iterative curriculum learning, this approach enables the detector to learn deep stylistic patterns rather than just superficial statistics.

3. **Strong Empirical Results.** The experiments demonstrate that I-PREF outperforms all statistical feature-based baselines. The method achieves high identification accuracy and proves particularly effective in the setting of distinguishing between closely-related family models (e.g., Claude-4 vs. Claude-3.5).

**Weaknesses:**

1. **Disconnect between Motivation and Experiments.** The paper's motivation (Hacking) and its experiments (Identification) are disconnected. The paper's goal is to "hack" LM Arena, but its methods and experiments are heavily focused on model response identification. It is unclear if a binary detector for a single target model can have a significant impact on Arena, especially given the low probability (~1%) of sampling any single target model from a pool of hundreds. While prior work (e.g., Min et al., 2025) proposed attack strategies, this paper does not demonstrate whether I-PREF can be effectively used within any similar strategies. It would be nice to add simulation experiments to demonstrate a quantifiable impact on Elo ratings with I-PRF.
2. **Incomplete Analysis.** Given the paper's claim to "hack" LM Arena, an analysis connecting the reported identification accuracy to the potential impact on Arena ratings is missing. Quantifying how an identification accuracy connects with Elo ratings would have made the paper's claims much stronger.
3. **Unrealistic Experimental Setup.** The authors selected 12 models with a wide range of capabilities, including 3B-sized models and frontier models, and selected only 3 top-performing LLMs as the target model. This significant capability gap may make the identification task artificially easy. A more realistic hacking scenario would involve selecting a set of ~10 models with very close Arena rating (e.g., within a $\pm 20$ Elo range) and testing the method. This would better reflect the real-world goal of making a model outperform its direct competitors.
4. **Weak Baselines.** Since the paper's experiments are primarily focused on model identification, or "source attribution," using only feature-based logistic models (e.g., TF-IDF, BoW) as baselines is insufficient. The authors should have included stronger baselines from the literature on model source attribution/identification, such as perplexity-based methods, neural-based classifiers, or LLM-as-a-judge baselines.

**Questions:**

1. According to Table 2, the identification accuracy between GPT-4o and GPT-4o-mini is only 67.5%, which is quite low compared to other pairs11. Could the authors provide more insights or analysis on why this is the case?
2. How does LM Arena's 'Style Control' feature, which is designed to mitigate stylistic biases, impact the effectiveness of the hacking method?

---

> ### Author Response · Authors · 2025-11-24
> **Response to Reviewer PG4o [1/5]**
>
> Dear Reviewer PG4o,
>
> We sincerely appreciate your thoughtful and constructive review. We provide detailed responses to each of your questions below:
>
> ---
>
> **[W1] Simulation experiments to demonstrate a quantifiable impact on Elo ratings with I-PREF**
>
> Thank you for the insightful comment. To directly address the concern regarding whether I-PREF can be effectively used within realistic vote-rigging strategies, we conducted new simulation experiments measuring the concrete impact an I-PREF–equipped adversary can exert on Elo-style rankings. Our simulation setup follows the framework of [1] specifically, we use arena-human-preference-140k dataset [2], and reconstruct the Arena environment using a Bradley–Terry (BT) based ranking model. At each step, an attacker interacts with the system, receives two anonymized model responses, and follows the same decision protocol described in our main setup. Full details of the simulation environment are provided in Appendix E.
>
> **Scenario #1: Passive “do-nothing” strategy when the target model is not sampled**
>
> We first analyze a passive scenario where the adversary acts only when the target model appears**,** reflecting the low sampling probability in Arena-like systems (3.3% in our simulation). When I-PREF detects the target, the adversary casts a strategic vote; otherwise, no action is taken. Despite the restrictive nature of this setting, reliable identification alone already enables meaningful manipulation of model rankings.
> As shown in the table below, demoting the top-ranked model from Rank 1 to Rank 2 requires only 11 adversarial downvotes, whereas promoting the second-ranked model from Rank 2 to Rank 1 requires 165 upvotes. This asymmetry demonstrates that, even with sparse opportunities, selective targeting guided by I-PREF produces measurable changes in Elo trajectories.
>
>
>  (A) Votes
>
> | Target model                     | Current rank | Target rank: 1 | Target rank: 2 | Target rank: 3 | Target rank: 4 | Target rank: 5 |
> |----------------------------------|--------------|----------------|----------------|----------------|----------------|----------------|
> | gemini-2.5-pro                   | 1            | N/A            | ↓ 11          | ↓ 409          | ↓ 507          | ↓ 856          |
> | gemini-2.5-pro-preview-03-25     | 2            | ↑ 165          | N/A            | ↓ 13           | ↓ 61           | ↓ 61           |
> | grok-4-0709                      | 3            | ↑ 218          | ↑ 25           | N/A            | ↓ 52           | ↓ 56           |
> | o3-2025-04-16                    | 4            | ↑ 1791         | ↑ 639          | ↑ 466          | N/A            | ↓ 5            |
> | chatgpt-4o-latest-20250326       | 5            | ↑ 1615         | ↑ 489          | ↑ 439          | ↑ 37           | N/A            |
>
>  (B) Interactions
>
> | Target model                     | Current rank | Target rank: 1 | Target rank: 2 | Target rank: 3 | Target rank: 4 | Target rank: 5 |
> |----------------------------------|--------------|----------------|----------------|----------------|----------------|----------------|
> | gemini-2.5-pro                   | 1            | N/A            | ↓ 200         | ↓ 12100        | ↓ 14900        | ↓ 24700        |
> | gemini-2.5-pro-preview-03-25     | 2            | ↑ 5000         | N/A            | ↓ 500          | ↓ 2000         | ↓ 2000         |
> | grok-4-0709                      | 3            | ↑ 6400         | ↑ 600          | N/A            | ↓ 1400         | ↓ 1500         |
> | o3-2025-04-16                    | 4            | ↑ 49300        | ↑ 18200        | ↑ 13400        | N/A            | ↓ 100          |
> | chatgpt-4o-latest-20250326       | 5            | ↑ 43800        | ↑ 13700        | ↑ 12400        | ↑ 1000         | N/A            |
>
>
> **Scenario #2: Aggressive manipulation by “downvoting high-ranking competitors”**
>
> We further examined a stronger and more realistic manipulation strategy, where the adversary leverages I-PREF not only to detect the target, but also to identify and strategically downvote a predefined set of high-ranking competitors (“enemy list”) even when the target is absent. The results below show a substantial improvement in manipulation efficiency. For instance, promoting o3-2025-04-16 from Rank 4 to Rank 1 required 49,300 interactions under the passive regime, but only 14,400 under the aggressive strategy, more than a 3× reduction. Similar gains appear across all top models.

---

> ### Author Response · Authors · 2025-11-24
> **Response to Reviewer PG4o [2/5]**
>
> (a) Votes
>
> | Target model                   | Current rank | Target rank: 1 | Target rank: 2 | Target rank: 3 | Target rank: 4 | Target rank: 5 |
> |--------------------------------|--------------|----------------|----------------|----------------|----------------|----------------|
> | gemini-2.5-pro                 | 1            | N/A            | ↓ 66           | ↓ 628          | ↓ 1060         | ↓ 1371         |
> | gemini-2.5-pro-preview-03-25   | 2            | ↑ 240          | N/A            | ↓ 34           | ↓ 34           | ↓ 169          |
> | grok-4-0709                    | 3            | ↑ 315          | ↑ 34           | N/A            | ↓ 56           | ↓ 168          |
> | o3-2025-04-16                  | 4            | ↑ 1188         | ↑ 250          | ↑ 225          | N/A            | ↓ 78           |
> | chatgpt-4o-latest-20250326     | 5            | ↑ 1169         | ↑ 244          | ↑ 196          | ↑ 64           | N/A            |
>
> (b) Interactions
>
> | Target model                   | Current rank | Target rank: 1 | Target rank: 2 | Target rank: 3 | Target rank: 4 | Target rank: 5 |
> |--------------------------------|--------------|----------------|----------------|----------------|----------------|----------------|
> | gemini-2.5-pro                 | 1            | N/A            | ↓ 1900         | ↓ 15400        | ↓ 24000        | ↓ 26800        |
> | gemini-2.5-pro-preview-03-25   | 2            | ↑ 3100         | N/A            | ↓ 500          | ↓ 500          | ↓ 1900         |
> | grok-4-0709                    | 3            | ↑ 4100         | ↑ 300          | N/A            | ↓ 500          | ↓ 1500         |
> | o3-2025-04-16                  | 4            | ↑ 14400        | ↑ 1900         | ↑ 1600         | N/A            | ↓ 500          |
> | chatgpt-4o-latest-20250326     | 5            | ↑ 14600        | ↑ 2000         | ↑ 1500         | ↑ 500          | N/A            |
>
> Overall, these findings confirm that I-PREF is not merely a response-identification tool but can be directly incorporated into practical vote-rigging strategies to produce quantifiable and substantial shifts in Elo-based rankings. This addresses the reviewer’s concern by empirically demonstrating that high-accuracy detection translates into meaningful manipulation capability in Arena-like environments. We have added these results and the corresponding discussion to Appendices E.2 and E.3 of the revised draft.
>
> ---
>
>  **[W2] Quantifying how an identification accuracy connects with Arena ranking**
>
> We acknowledge that connecting identification accuracy to its impact on Arena rankings is important. To directly address this gap, we conducted additional simulations quantifying how detectors with different accuracy levels influence the number of Arena interactions required to shift Elo-based rankings. For the experiments, we use the same setups of experiments in [W1]. The results are summarized below:
>
>
> (a) Votes
>
> | Detector        | Target rank: 1 (↑ 4) | Target rank: 2 (↑ 3) | Target rank: 3 (↑ 2) | Target rank: 4 (↑ 1) |
> |-----------------|-----------------------|-----------------------|-----------------------|-----------------------|
> | LENGTH-CHAR     | ↑ 1619                | ↑ 538                 | ↑ 404                 | ↑ 34                  |
> | LENGTH-WORD     | ↑ 1616                | ↑ 560                 | ↑ 394                 | ↑ 36                  |
> | TF-IDF          | ↑ 1630                | ↑ 563                 | ↑ 394                 | ↑ 36                  |
> | Neural-Based    | ↑ 1604                | ↑ 526                 | ↑ 448                 | ↑ 36                  |
> | BoW             | ↑ 1631                | ↑ 562                 | ↑ 383                 | ↑ 35                  |
> | LLM-judge       | ↑ 1619                | ↑ 575                 | ↑ 459                 | ↑ 34                  |
> | I-PREF (Ours) | ↑ 1618            | ↑ 564             | ↑ 442             | ↑ 37              |

---

> ### Author Response · Authors · 2025-11-24
> **Response to Reviewer PG4o [3/5]**
>
> (b) Interactions
>
> | Detector        | Target rank: 1 (↑ 4) | Target rank: 2 (↑ 3) | Target rank: 3 (↑ 2) | Target rank: 4 (↑ 1) |
> |-----------------|-----------------------|-----------------------|-----------------------|-----------------------|
> | LENGTH-CHAR     | ↑ 73600               | ↑ 24600               | ↑ 18200               | ↑ 1700                |
> | LENGTH-WORD     | ↑ 64500               | ↑ 21400               | ↑ 14600               | ↑ 1300                |
> | TF-IDF          | ↑ 55100               | ↑ 18500               | ↑ 12400               | ↑ 1100                |
> | Neural-Based    | ↑ 54500               | ↑ 18300               | ↑ 15900               | ↑ 1200                |
> | BoW             | ↑ 52300               | ↑ 17700               | ↑ 11600               | ↑ 1000                |
> | LLM-judge       | ↑ 63800               | ↑ 22200               | ↑ 18300               | ↑ 1500                |
> | I-PREF (Ours) | ↑ 45800          | ↑ 16800           | ↑ 13000         | ↑ 1000            |
>
> The results clearly show that detectors with lower identification accuracy (e.g., LENGTH-CHAR (58.6%) or LLM-as-a-judge baselines (64.3%) require substantially more Arena interactions to promote a target model. Although the number of adversarial votes required is similar across baselines, the total number of Arena interactions varies significantly (45.8k–73.6k). This discrepancy arises because low-accuracy methods frequently misidentify models, causing attackers to waste interactions on uninformative or counterproductive comparisons, thereby inflating the overall attack cost.
>
> In contrast, I-PREF not only achieves the highest identification accuracy but also provides the most efficient mechanism for influencing leaderboard dynamics under limited interaction budgets. These findings directly support the paper’s claim regarding the vulnerability of LM Arena’s ranking mechanism under the strong identification method like I-PREF. We have incorporated these results and the corresponding discussion into Appendices E.1 and E.2 of the revised draft.
>
> ---
>
> **[W3] A more realistic hacking scenario by selecting a set of ~10 models with very close Arena rating (e.g., within an Elo range)**
>
> We appreciate the reviewer’s insightful comment. However, we would first like to clarify that Elo rating alone is not a reliable proxy for identification difficulty. As shown in Table 2 of our main results, models with substantially different Elo scores can still exhibit similar stylistic or structural features, making the discrimination task non-trivial (e.g., GPT-4o vs. GPT-4o-mini). Conversely, some models with very close Elo ratings remain highly distinguishable (e.g., GPT-4o vs. Gemini-Pro). This result empirically supports our claim that Elo rating does not directly predict the hardness of the identification task.
>
> Nevertheless, we agree that demonstrating effectiveness under the reviewer’s suggested scenario would further strengthen our claims. To this end, we additionally conducted a new experiment using a group of models with highly similar Arena Elo ratings. The numbers in parentheses below indicate each model’s LM Arena Elo rating as reported in the official leaderboard (captured on 25-11-21). Specifically, we selected:
>
> • glm-4.6 (1427), qwen3-max-0923 (1423), grok-4-fast (1420), deepseek-v3.1 (1416), kimi-k2-0905 (1417), chatgpt-4o (1438), claude-sonnet-4 (1389), gemini-2.5-pro (1451)
>
> Following the setup in Section 4, we sampled 1,400 prompts, collected responses from all eight models, and trained and evaluated I-PREF under this constrained “close-Elo” setting. As in the main paper, we consider GPT-4o, Gemini-Pro, and Claude-4 as the target LLMs.
>
>
> | Target      | GPT-4o | Gemini-Pro | Claude-4 | grok-4-fast | DeepSeek-v3.1 | glm-4.6 | kimi-k2 | qwen3-max |
> |-------------|--------|------------|----------|-------------|----------------|---------|---------|-----------|
> | GPT-4o      | --     | 0.985      | 0.975    | 0.935       | 0.950          | 0.975   | 0.950   | 0.875     |
> | Gemini-Pro  | 0.885  | --         | 0.890    | 0.935       | 0.675          | 0.565   | 0.835   | 0.910     |
> | Claude-4    | 0.975  | 1.000      | --       | 0.985       | 0.995          | 0.995   | 0.990   | 0.980     |
>
>
> As the table shows, I-PREF maintains strong and stable identification performance even among models with nearly identical Elo ratings. These findings demonstrate that our method does not rely on trivial capability gaps; rather, it learns deep, model-specific stylistic and reasoning patterns that persist even among direct competitors on the Arena leaderboard. We include these additional experiments and the corresponding analysis in Appendix D.5 of the revised draft.

---

> ### Author Response · Authors · 2025-11-24
> **Response to Reviewer PG4o [4/5]**
>
> **[W4] Stronger baselines from the literature on model source attribution/identification, such as perplexity-based methods, neural-based classifiers, or LLM-as-a-judge baselines**
>
> Thank you for pointing out the missing baselines. To address this concern, we have incorporated substantially stronger baselines for model source identification beyond simple statistical methods. Specifically, we implemented (1) a neural multi-class transformer classifier trained to attribute each response to one of the candidate models [3], and (2) a strong LLM-as-a-judge baseline equipped with few-shot examples of identification cases. Detailed implementation descriptions are provided in Appendix D.6, and the corresponding results are summarized below:
>
> | Method        | GPT-4o | Gemini-Pro | Claude-4 | Average |
> |---------------|--------|------------|----------|---------|
> | LLM-judge     | 0.679  | 0.620      | 0.631    | 0.643   |
> | Neural-Based  | 0.726  | 0.828      | 0.825    | 0.793   |
> | I-PREF (Ours) | 0.908 | 1.000 | 0.946 | 0.951 |
>
> As shown in the table, both the neural classifier and the LLM-as-a-judge baseline struggle to reliably discriminate among high-performing models such as GPT-4o, Gemini-Pro, and Claude-4, achieving accuracies in the 0.62–0.83 range. This reflects the inherent difficulty of the identification task due to the high stylistic and structural similarity across modern frontier LLMs.
> In contrast, I-PREF achieves significantly higher accuracy across all target models (0.908–1.000), with an average accuracy of 0.951, outperforming both neural and LLM-based attribution methods by a large margin. These results demonstrate that even when compared against significantly stronger and more sophisticated baselines, I-PREF provides the most reliable and consistent source-identification performance. We will incorporate these new baselines into the main tables of the final draft.
>
> ---
>
>  **[Q1] According to Table 2, the identification accuracy between GPT-4o and GPT-4o-mini is only 67.5%, which is quite low compared to other pairs11. Could the authors provide more insights or analysis on why this is the case?**
>
> The relatively lower accuracy between GPT-4o and GPT-4o-mini is expected because this pair represents the closest relationship among all models in our study. GPT-4o-mini is a compact model explicitly designed to receive distillation from GPT-4o [4] enabling it to reproduce similar behaviors at lower cost and latency. This distillation-compatible design leads the two systems to share highly similar stylistic, lexical, and structural generation patterns, resulting in minimal divergence and making this pair intrinsically the most challenging identification scenario.
>
> In contrast, other within-family comparisons—such as Gemini-Pro vs. Gemini-Flash or Claude-3.5 vs. Claude-4—show relatively larger differences in both style and behavior. Gemini-Flash follows a design optimized for speed and efficiency [5], and Claude-3.5 and Claude-4 come from different model generations with clear shifts in reasoning characteristics and phrasing patterns. Because these pairs exhibit broader divergence in their generation profiles, the detector has more discriminative cues to rely on, resulting in higher identification accuracy than for the tightly related GPT-4o / GPT-4o-mini pair.
>
> ---
>
> **[Q2] How does LM Arena's 'Style Control' feature, which is designed to mitigate stylistic biases, impact the effectiveness of the hacking method?**
>
> LM Arena’s Style Control is designed to debias leaderboard scoring by removing the influence of surface-level textual attributes [6,7] (e.g., response length, markdown formatting) after votes are collected. Namely, if some patterns could be easily identified from the response, their effect could be explicitly mitigated during ranking calculation. Therefore, this ‘Style Control’ feature would directly reduce the effectiveness of hacking methods based on surface-level textual attributes.
>
> However, this ‘Style Control’ feature is not sufficient to fully address the risk of hacking methods including our framework. Style Control operates only on pre-defined, human-interpretable stylistic signals, while I-PREF relies on deeper, model-driven generation characteristics that are not directly observable or removable through simple normalization rules. These signals emerge from the target model’s underlying generation dynamics rather than its superficial style, and thus remain intact even after stylistic debiasing.
>
> Moreover, prior statistical baselines that depend heavily on length, formatting, or shallow distributional cues are indeed neutralized by Style Control, but I-PREF consistently outperforms these baselines precisely because it does not depend on such brittle features. In short, Style Control mitigates only the simplest forms of manipulation, whereas I-PREF exploits features that cannot be trivially removed without altering the model’s underlying behavior.

---

> ### Author Response · Authors · 2025-11-24
> **Response to Reviewer PG4o [5/5]**
>
> ---
>
> [1] Huang et al., Exploring and Mitigating Adversarial Manipulation of Voting-Based Leaderboards, ICML 2025
> [2] https://huggingface.co/datasets/lmarena-ai/arena-human-preference-140k
> [3] Yi Tay et al., Reverse Engineering Configurations of Neural Text Generation Models, ACL 2020
> [4] https://platform.openai.com/docs/models/gpt-4o-mini
> [5] https://developers.googleblog.com/en/gemini-2-family-expands/
> [6] https://lmsys.org/blog/2024-08-28-style-control/
> [7] https://colab.research.google.com/drive/19VPOril2FjCX34lJoo7qn4r6adgKLioY?ref=news.lmarena.ai#scrollTo=C4xnVybEy0OO
>
> ---
>
> We hope our responses resolve the points you raised. If you have any further questions, please don't hesitate and let us know.
>
> Thank you very much,
> Authors

---

### Official Review · Reviewer_fTGT · 2025-11-01

**Soundness:** 3
**Presentation:** 4
**Contribution:** 2
**Rating:** 6
**Confidence:** 2

**Summary:**

This paper proposes I-PREF, a model-driven framework for LLM identification leveraging interpolated preference learning. The core idea is to train a detector to distinguish a target LLM’s responses from others by creating synthetic hard negatives via model interpolation, and to improve identification performance via adaptive and iterative curriculum learning. Experimental results show that I-PREF markedly outperforms feature-based identification baselines in accuracy and AUROC across both standardized and crowdsourced human preference datasets

**Strengths:**

- The work introduces a novel preference learning methodology that demonstrates superior performance and considerable robustness compared to traditional feature-based baselines like TF-IDF and BoW.

- The manuscript features excellent presentation and writing quality, coupled with clear experimental design, resulting in a substantial and rich content.

-  The experimental design, including comprehensive ablation studies and direct comparisons against advanced current baselines, is well-executed and provides significant empirical insights.

**Weaknesses:**

- Limited Scalability Analysis for Large Models: The study lacks experimentation on substantially larger model scales. Current experiments are confined to models below 10 billion parameters (<10B), omitting a thorough analysis for models exceeding 30 billion parameters (>30B). It remains unclear whether I-PREF maintains comparable efficacy when applied to LLMs possessing greater capabilities and scale.

**Questions:**

same as weakness

---

> ### Author Response · Authors · 2025-11-24
> **Response to Reviewer fTGT**
>
> Dear Reviewer fTGT,
>
> We sincerely appreciate your thoughtful and constructive review. We provide detailed responses to each of your questions below:
>
> ---
>
> **[W1]  Limited Scalability Analysis for Large Models**
>
>
> Thank you for the thoughtful comments. To directly address the scalability concern, we conducted new experiments by scaling up our reward model from 3B to 27B parameters. The results clearly demonstrate that I-PREF remains effective even when integrated with a substantially larger reward model, extending beyond the <10B setting presented in the main paper.
> Specifically, we replaced the 3B-scale reward model used in the main experiments with the 27B QRM-Gemma-2-27B model [1] and retrained all components of I-PREF (doublet, triplet, and iterative). This setup allows us to evaluate whether preference learning with interpolated hard negatives continues to behave reliably under significantly higher-capacity reward models. The results are summarized below:
>
>
> | Target LLM   | Accuracy (Pairwise) | Accuracy (Triplet) | Accuracy (Iterative) | AUROC (Pairwise) | AUROC (Triplet) | AUROC (Iterative) |
> |--------------|----------------------|----------------------|------------------------|-------------------|------------------|---------------------|
> | GPT-4o       | 0.7197               | 0.7522               | 0.8055                 | 0.7252            | 0.7256           | 0.7718              |
> | Gemini-Pro   | 0.8375               | 0.8618               | 0.9350                 | 0.8241            | 0.8521           | 0.8947              |
> | Claude-4     | 0.8545               | 0.8650               | 0.8868                 | 0.8713            | 0.8722           | 0.8746              |
>
> Across all three target LLMs (GPT-4o, Gemini-Pro, Claude-4), we observe consistent improvements in both AUROC and Accuracy when using the 27B reward model. These gains appear across all configurations (doublet, triplet, and iterative) indicating that increased reward-model capacity further strengthens the preference-learning pipeline. We have added these results and the corresponding discussion to Appendix D.1 of the revised draft.
>
>
> For completeness, we would also like to highlight additional experiments in Appendix D.5, where we evaluate I-PREF against even more challenging comparison models with substantially higher Elo ratings (e.g., qwen3-max-0923, which exceeds 1T parameters). Even in this setting, I-PREF continues to reliably identify the target LLM, further validating its scalability and robustness.
>
> ---
>
> [1] QRM-Gemma-2-27B: https://huggingface.co/nicolinho/QRM-Gemma-2-27B
>
> ---
>
> If you have any further questions, please let us know.
>
> Thank you very much,
> Authors

---

### Author Response · Authors · 2025-11-24
**General Response**

Dear Reviewers and AC,


We sincerely appreciate your valuable time and effort spent reviewing our manuscript.
As reviewers highlighted, I-PREF is a conceptually **simple yet genuinely novel preference-learning framework (fTGT, xXHV)** that leverages economical interpolated hard-negative synthesis and triplet-based optimization to capture model-specific behavioral cues beyond surface features. The method remains **practical and data-efficient (QwZn, xXHV)**, and delivers **strong, consistent empirical gains (all reviewers).**


We appreciate your constructive and valuable feedback on our manuscript. We have carefully updated and improved the manuscript in response to the feedbacks , including the following additional discussions and experiments:


* Clarification on threat model (Lines 171-180 in Section 3)
* Alternative reward model architectures  (Appendix D.1)
* Effect on copymodel capacity on identification performance (Appendix D.2)
* Evaluation on open-source target models  (Appendix D.3)
* Effect of interpolation strength(α)  (Appendix D.4)
* Identification on models with similar Elo ratings  (Appendix D.5)
* Additional baselines for comparison (Appendix D.6)
* Simulation for ranking manipulation (Appendix E)

In the revised manuscript, these updates are temporarily highlighted in $\textbf{\color{green}green}$ for your convenience to check.

We sincerely believe that these updates may help us better deliver the benefits of the proposed I-PREF to the ICLR community.


Sincerely,


Authors.

---

### Author Response · Authors · 2025-11-28
**A Gentle Invitation for Further Discussion**

Dear reviewers,

Once again, we would like to kindly express our appreciation for your thoughtful reviews and the time you have devoted to evaluating our submission.

If you have any additional thoughts or follow-up comments after reading our author response, we would be very grateful to hear them. Your insights during the discussion phase are extremely valuable and help us further clarify and strengthen the work.

Thank you very much for your time and engagement.

Kindest regards,
Authors

---

### Author Response · Authors · 2025-12-03
**Summary of Rebuttal Phase**

Dear Area Chair,

We sincerely appreciate your time and effort in coordinating the review process and supporting the ICLR community.


While the reviewers acknowledged the novelty and advantages of our work as we summarized in the general response, their initial assessments were divided. During the discussion period, we submitted a detailed rebuttal and revised draft addressing all concerns raised. **However, only one reviewer (QwZn) participated in the discussion**, likely due to the unexpectedly early closing of the discussion phase, leaving the remaining reviewers unable to reconsider their initial judgments.

**Despite the limited participation, the available evidence suggests that our responses effectively resolved the main concerns.** Reviewer QwZn explicitly stated that the new experiments and explanations fully addressed the concerns and **updated their score from 4 → 6 before reverting** (“The added experiment and explanation addressed my concerns and I have updated my score accordingly.”). Given that QwZn shared concerns similar to those of the other reviewers, we believe this provides a meaningful indication of how the remaining reviewers might have updated their evaluations had they been able to participate.

Below, we briefly summarize the concerns from the non-participating reviewers and how the rebuttal and revised draft addresses them:

* **Concerns from Reviewers PG4o and xXHV regarding the practical impact of identification accuracy**

Both reviewers questioned whether I-PREF’s identification capability translates into real influence under vote-rigging scenarios. In response, we added new Arena-style simulations showing (1) that even with low appearance probability (~3%), reliable identification already produces measurable Elo shifts, and (2) that under realistic aggressive strategies, where adversaries also downvote high-ranking competitors—, The required number of interactions decreases by more than 3×. These results demonstrate that I-PREF can meaningfully influence rankings when incorporated into practical manipulation strategies.

* **Concerns regarding stronger baselines and experimental setups by Reviewer PG4o**

We added new experiments including neural classifiers and LLM-as-a-judge baselines, which were substantially stronger than the original feature-based baselines. I-PREF outperformed all of them by a large margin. Reviewer QwZn found these additions sufficient to resolve the same concern.

* **Additional analyses requested by Reviewer xXHV**

We incorporated experiments involving open-source substitutes, interpolation-strength sensitivity, and a broader model-scale analysis. These demonstrate that I-PREF behaves robustly across interpolation strengths, reward-model sizes, and both proprietary and open-source targets.

---
Overall, although PG4o and xXHV did not participate in discussion, we believe that all of their concerns were fully addressed in the revised draft. We are confident that our clarified analyses and expanded experiments satisfactorily resolve the issues they raised.


Again, we sincerely appreciate the reviewers’ valuable feedback throughout the process and believe that the revised paper is now substantially strengthened and will be of significant interest to the ICLR audience.


Thank you again for your thoughtful consideration.

Best regards,

Authors

---

### Note · Program_Chairs · 2026-01-17
**Submission Desk Rejected by Program Chairs**

The following references in this submission do not refer to real documents and/or have major errors in bibliographic information:

 Tuan Anh Le and et al. Adversarial examples for large language models. arXiv preprint arXiv:2010.00906, 2020.